# EM Distillation for One-step Diffusion Models

**Sirui Xie**[1,2,3]  **Zhisheng Xiao**[2]  **Diederik P. Kingma**[1]  **Tingbo Hou**[2]
**Ying Nian Wu**[3]  **Kevin Murphy**[1]  **Tim Salimans**[1]  **Ben Poole**[1]  **Ruiqi Gao**[1]
[1]Google DeepMind    [2]Google Research    [3]UCLA

## Abstract

While diffusion models can learn complex distributions, sampling requires a computationally expensive iterative process. Existing distillation methods enable efficient sampling, but have notable limitations, such as performance degradation with very few sampling steps, reliance on training data access, or mode-seeking optimization that may fail to capture the full distribution. We propose EM Distillation (EMD), a maximum likelihood-based approach that distills a diffusion model to a one-step generator model with minimal loss of perceptual quality. Our approach is derived through the lens of Expectation-Maximization (EM), where the generator parameters are updated using samples from the joint distribution of the diffusion teacher prior and inferred generator latents. We develop a reparametrized sampling scheme and a noise cancellation technique that together stabilize the distillation process. We further reveal an interesting connection of our method with existing methods that minimize mode-seeking KL. EMD outperforms existing one-step generative methods in terms of FID scores on ImageNet-64 and ImageNet-128, and compares favorably with prior work on distilling text-to-image diffusion models.

## 1  Introduction

Diffusion models [1–3] have enabled high-quality generation of images [4–6], videos [7, 8], and other modalities [9–11]. Diffusion models use a forward process to create a sequence of distributions that transform the complex data distribution into a Gaussian distribution, and learn the score function for each of these intermediate distributions. Sampling from a diffusion model reverses this forward process to create data from random noise by solving an SDE, or an equivalent probability flow ODE [12]. Typically, solving this differential equation requires a significant number of evaluations of the score function, resulting in a high computational cost. Reducing this cost to single function evaluation would enable applications in real-time generation.

To enable efficient sampling from diffusion models, two distinct approaches have emerged: (1) trajectory distillation methods [13–18] that accelerate solving the differential equation, and (2) distribution matching approaches [19–23] that learn implicit generators to match the marginals learned by the diffusion model. Trajectory distillation-based approaches have greatly reduced the number of steps required to produce samples, but continue to face challenges in the 1-step generation regime. Distribution matching approaches can enable the use of arbitrary generators and produce more compelling results in the 1-step regime, but often fail to capture the full distribution due to the *mode-seeking* nature of the divergences they minimize.

In this paper, we propose EM Distillation (EMD), a diffusion distillation method that minimizes an approximation of the *mode-covering* divergence between a pre-trained diffusion teacher model and a latent-variable student model. The student enables efficient generation by mapping from noise to data in just one step. To achieve Maximum Likelihood Estimation (MLE) of the marginal teacher distribution for the student, we propose a method similar to the Expectation-Maximization (EM) framework [24], which alternates between an Expectation-step (E-step) that estimates the learning gradients with Monte Carlo samples, and a Maximization-step (M-step) that updates the student

through gradient ascent. As the target distribution is represented by the pre-trained score function, the E-step in the original EM that first samples a datapoint and then infers its implied latent variable would be expensive. We introduce an alternative MCMC sampling scheme that jointly updates the data and latent pairs initialized from student samples, and develop a reparameterized approach that simplifies hyperparameter tuning and improves performance for short-run MCMC [25]. For the optimization in the M-step given these joint samples, we discover a tractable linear noise term in the learning gradient, whose removal significantly reduces variances. Additionally, we identify a connection to Variational Score Distillation [9, 26] and Diff-Instruct [22], and show how the strength of the MCMC sampling scheme can interpolate between mode-seeking and mode-covering divergences. Empirically, we first demonstrate that a special case of EMD, which is equivalent to the Diff-Instruct [22] baseline, can be readily scaled and improved to achieve strong performance. We further show that the general formulation of EMD that leverages multi-step MCMC can achieve even more competitive results. For ImageNet-64 and ImageNet-128 conditional generation, EMD outperforms existing one-step generation approaches with FID scores of 2.20 and 6.0. EMD also performs favorably on one-step text-to-image generation by distilling from Stable Diffusion models.

## 2 Preliminary

### 2.1 Diffusion models and score matching

Diffusion models [1, 2], also known as score-based generative models [27, 3], consist of a forward process that gradually injects noise to the data distribution and a reverse process that progressively denoises the observations to recover the original data distribution $p_{\text{data}}(\mathbf{x}_0)$. This results in a sequence of noise levels $t \in (0, 1]$ with conditional distributions $q_t(\mathbf{x}_t|\mathbf{x}_0) = \mathcal{N}(\alpha_t \mathbf{x}_0, \sigma_t^2 \boldsymbol{I})$, whose marginals are $q_t(\mathbf{x}_t)$. We use a variance-preserving forward process [3, 28, 29] such that $\sigma_t^2 = 1 - \alpha_t^2$. Song et al. [3] showed that the reverse process can be simulated with a reverse-time Stochastic Differential Equation (SDE) that depends only on the time-dependent score function $\nabla_{\mathbf{x}_t} \log p_t(\mathbf{x}_t)$ of the marginal distribution of the noisy observations. This score function can be estimated by a neural network $s_{\boldsymbol{\phi}}(\mathbf{x}_t, t)$ through (weighted) denoising score matching [30, 31]:

$$\mathcal{J}(\boldsymbol{\phi}) = \mathbb{E}_{p_{\text{data}}(\mathbf{x}_0), p(t), q_t(\mathbf{x}_t|\mathbf{x}_0)} \left[ w(t) \| s_{\boldsymbol{\phi}}(\mathbf{x}_t, t) - \nabla_{\mathbf{x}_t} \log q_t(\mathbf{x}_t|\mathbf{x}_0) \|_2^2 \right], \tag{1}$$

where $w(t)$ is the weighting function and $p(t)$ is the noise schedule.

### 2.2 MCMC with Langevin dynamics

While solving the reverse-time SDE results in a sampling process that traverses noise levels, simulating Langevin dynamics [32] results in a sampler that converges to and remains at the data manifold of a target distribution. As a particularly useful Markov Chain Monte Carlo (MCMC) sampling method for continuous random variables, Langevin dynamics generate samples from a target distribution $\rho(\mathbf{x})$ by iterating through

$$\mathbf{x}^{i+1} = \mathbf{x}^i + \gamma \nabla_{\mathbf{x}} \log \rho(\mathbf{x}^i) + \sqrt{2\gamma} \mathbf{n}, \tag{2}$$

where $\gamma$ is the stepsize, $\mathbf{n} \sim \mathcal{N}(\mathbf{0}, \boldsymbol{I})$, and $i$ indexes the sampling timestep. Langevin dynamics has been widely adopted for sampling from diffusion models [27, 3] and energy-based models [33–36]. Convergence of Langevin dynamics requires a large number of sampling steps, especially for high-dimensional data. In practice, short-run variants with early termination have been succesfully used for learning of EBMs [25, 37, 38].

### 2.3 Maximum Likelihood and Expectation-Maximization

Expectation-Maximization (EM) [24] is a maximum likelihood estimation framework to learn latent variable models: $p_{\boldsymbol{\theta}}(\mathbf{x}, \mathbf{z}) = p_{\boldsymbol{\theta}}(\mathbf{x}|\mathbf{z})p(\mathbf{z})$, such that the marginal distribution $p_{\boldsymbol{\theta}}(\mathbf{x}) = \int p_{\boldsymbol{\theta}}(\mathbf{x}, \mathbf{z})d\mathbf{z}$ approximates the target distribution $q(\mathbf{x})$. It originates from the generic training objective of *maximizing* the log-likelihood function over parameters: $\mathcal{L}(\boldsymbol{\theta}) = \mathbb{E}_{q(\mathbf{x})}[\log p_{\boldsymbol{\theta}}(\mathbf{x})]$, which is equivalent to *minimizing* the *forward* KL divergence $D_{\text{KL}}(q(\mathbf{x})||p_{\boldsymbol{\theta}}(\mathbf{x}))$ [39]. Since the marginal distribution $p_{\boldsymbol{\theta}}(\mathbf{x})$ is usually analytically intractable, EM involves an E-step that expresses the gradients over the model parameters $\theta$ with an expectation formula

$$\nabla_{\boldsymbol{\theta}} \mathcal{L}(\boldsymbol{\theta}) = \nabla_{\boldsymbol{\theta}} \mathbb{E}_{q(\mathbf{x})}[\log p_{\boldsymbol{\theta}}(\mathbf{x})] = \mathbb{E}_{q(\mathbf{x})p_{\boldsymbol{\theta}}(\mathbf{z}|\mathbf{x})}[\nabla_{\boldsymbol{\theta}} \log p_{\boldsymbol{\theta}}(\mathbf{x}, \mathbf{z})], \tag{3}$$

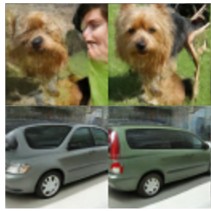 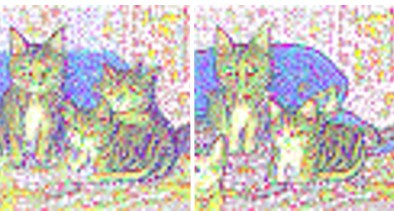 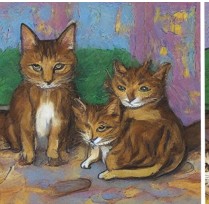 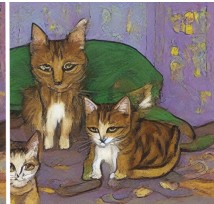

(a) ImageNet       (b) Text-to-image (SD embedding space)       (c) Text-to-image (SD image space)

Figure 1: **Before and after MCMC correction.** In (a)(b), the left columns are $\mathbf{x} = g_{\boldsymbol{\theta}}(\mathbf{z})$, the right columns are updated $\mathbf{x}$ after 300 steps of MCMC sampling jointly on $\mathbf{x}$ and $\mathbf{z}$. (a) illustrates the effect of correction in ImageNet. Note that the off-manifold images are corrected. (b) illustrates the correction in the embedding space of Stable Diffusion v1.5, which are decoded to image space in (c). Note the disentanglement of the cats and sharpness of the sofa. Zoom in for better viewing.

where $p_{\boldsymbol{\theta}}(\mathbf{z}|\mathbf{x}) = \frac{p_{\boldsymbol{\theta}}(\mathbf{x}|\mathbf{z})p(\mathbf{z})}{p_{\boldsymbol{\theta}}(\mathbf{x})}$ is the posterior distribution of $\mathbf{z}$ given $\mathbf{x}$. See Appendix A for a detailed derivation. The expectation can be approximated by Monte Carlo samples drawn from the posterior using e.g. MCMC sampling techniques. The estimated gradients are then used in an M-step to optimize the parameters. Han et al. [40] learned generator networks with an instantiation of this EM framework where E-steps leverage Langevin dynamics for drawing samples.

## 2.4 Variational Score Distillation and Diff-Instruct

Our method is also closely related to Score Distillation Sampling (SDS) [9], Variational Score Distillation (VSD) [26] and Diff-Instruct [22], which have been used for distilling diffusion models into a single-step generator [23, 41]. The generator produces clean images $\mathbf{x}_0 = g_{\boldsymbol{\theta}}(\mathbf{z})$ with $p(\mathbf{z}) = \mathcal{N}(\mathbf{0}, \boldsymbol{I})$, and can be diffused to noise level $t$ to form a latent variable model $p_{\boldsymbol{\theta},t}(\mathbf{x}_t, \mathbf{z}) = p_{\boldsymbol{\theta},t}(\mathbf{x}_t|\mathbf{z})p(\mathbf{z})$, $p_{\boldsymbol{\theta},t}(\mathbf{x}_t|\mathbf{z}) = \mathcal{N}(\alpha_t g_{\boldsymbol{\theta}}(\mathbf{z}), \sigma_t^2 \boldsymbol{I})$. This model is trained to match the marginal distributions $p_{\boldsymbol{\theta},t}(\mathbf{x}_t)$ and $q_t(\mathbf{x}_t)$ by minimizing their *reverse* KL divergence. Integrating over all noise levels, the objective is to *minimize* $\mathcal{J}(\boldsymbol{\theta})$ where

$$\mathcal{J}(\boldsymbol{\theta}) = \mathbb{E}_{p(t)}[\tilde{w}(t)D_{\mathrm{KL}}(p_{\boldsymbol{\theta},t}(\mathbf{x}_t)||q_t(\mathbf{x}_t))] = \mathbb{E}_{p(t)}\left[\tilde{w}(t)\int p_{\boldsymbol{\theta},t}(\mathbf{x}_t)\log\frac{p_{\boldsymbol{\theta},t}(\mathbf{x}_t)}{q_t(\mathbf{x}_t)}d\mathbf{x}_t\right]. \quad (4)$$

When parametrizing $\mathbf{x}_t = \alpha_t g_{\boldsymbol{\theta}}(\mathbf{z}) + \sigma_t \boldsymbol{\epsilon}$, the gradient for this objective in Eq. (4) can be written as

$$\nabla_{\boldsymbol{\theta}}\mathcal{J}(\boldsymbol{\theta}) = \mathbb{E}_{p(t),p(\boldsymbol{\epsilon}),p(\mathbf{z})}[-\tilde{w}(t)(\underbrace{\nabla_{\mathbf{x}_t}\log q_t(\mathbf{x}_t)}_{\text{teacher score}} - \underbrace{\nabla_{\mathbf{x}_t}\log p_{\boldsymbol{\theta},t}(\mathbf{x}_t)}_{\text{learned } s_\phi(\mathbf{x}_t,t)})\alpha_t\nabla_{\boldsymbol{\theta}}g_{\boldsymbol{\theta}}(\mathbf{z})], \quad (5)$$

where $p(\boldsymbol{\epsilon}) = \mathcal{N}(\mathbf{0}, \boldsymbol{I})$, the teacher score is provided by the pre-trained diffusion model. In SDS, $\nabla_{\mathbf{x}_t}\log p_{\boldsymbol{\theta},t}(\mathbf{x}_t)$ is the known analytic score function of the Gaussian generator. In VSD and Diff-Instruct, an auxiliary score network $s_\phi(\mathbf{x}_t, t)$ is learned to estimate it. The training alternates between learning the generator network $g_{\boldsymbol{\theta}}$ with the gradient update in Eq. (5) and learning the score network $s_\phi$ with the denoising score matching loss in Eq. (1).

## 3 Method

### 3.1 EM Distillation

We consider formulating the problem of distilling a pre-trained diffusion model to a deep latent-variable model $p_{\boldsymbol{\theta},t}(\mathbf{x}_t, \mathbf{z})$ defined in Section 2.4 using the EM framework introduced in Section 2.3. For simplicity, we begin with discussing the framework at a single noise level and drop the subscript $t$. We will revisit the integration over all noise levels in Section 3.3. Assume the target distribution $q(\mathbf{x})$ is represented by the diffusion model where we can access the score function $\nabla_{\mathbf{x}}\log q(\mathbf{x})$. Theoretically speaking, the generator network $g_{\boldsymbol{\theta}}(\mathbf{z})$ can employ any architecture including ones where the dimensionality of the latents differs from the data dimensionality. In this work, we reuse the diffusion denoiser parameterization as in other work on one-step distillation: $g_{\boldsymbol{\theta}}(\mathbf{z}) = \hat{\mathbf{x}}_{\boldsymbol{\theta}}(\mathbf{z}, t^*)$, where $\hat{\mathbf{x}}_{\boldsymbol{\theta}}$ is the $\mathbf{x}$-prediction function inherited from the teacher diffusion model, and $t^*$ remains a hyper-parameter.

A naive implementation of the E-step involves two steps: (1) draw samples from the target diffusion model $q(\mathbf{x})$ and (2) sample the latent variable $\mathbf{z}$ from $p_{\boldsymbol{\theta}}(\mathbf{z}|\mathbf{x})$ with *e.g.* MCMC techniques. Both steps can be highly non-trivial and computationally expensive, so here we present an alternative approach to sampling the same target distribution that avoids directly sampling from the pretrained diffusion model, by instead running MCMC from the joint distribution of $(\mathbf{x}, \mathbf{z})$. We initialize this sampling process using a joint sample from the student: drawing $\mathbf{z} \sim p(\mathbf{z})$ and $\mathbf{x} \sim p_{\boldsymbol{\theta}}(\mathbf{x}|\mathbf{z})$. This sampled $\mathbf{x}$ is no longer drawn from $q(\mathbf{x})$, but $\mathbf{z}$ is guaranteed to be a valid sample from the posterior $p_{\boldsymbol{\theta}}(\mathbf{z}|\mathbf{x})$. We then run MCMC to correct the sampled pair towards the desired distribution: $\rho_{\boldsymbol{\theta}}(\mathbf{x}, \mathbf{z}) := q(\mathbf{x})p_{\boldsymbol{\theta}}(\mathbf{z}|\mathbf{x}) = p_{\boldsymbol{\theta}}(\mathbf{x}, \mathbf{z})\frac{q(\mathbf{x})}{p_{\boldsymbol{\theta}}(\mathbf{x})}$ (see Fig. 1 for a visualization of this process). If $q(\mathbf{x})$ and $p_{\boldsymbol{\theta}}(\mathbf{x})$ are close to each other, $\rho_{\boldsymbol{\theta}}(\mathbf{x}, \mathbf{z})$ is close to $p_{\boldsymbol{\theta}}(\mathbf{x}, \mathbf{z})$. In that case, initializing the *joint* sampling of $\rho_{\boldsymbol{\theta}}(\mathbf{x}, \mathbf{z})$ with pairs of $(\mathbf{x}, \mathbf{z})$ from $p_{\boldsymbol{\theta}}(\mathbf{x}, \mathbf{z})$ could significantly accelerate both sampling of $\mathbf{x}$ and inference of $\mathbf{z}$. Assuming MCMC converges, we can use the resulting samples to estimate the learning gradients for EM:

$$\nabla_{\boldsymbol{\theta}}\mathcal{L}(\boldsymbol{\theta}) = \mathbb{E}_{\rho_{\boldsymbol{\theta}}(\mathbf{x},\mathbf{z})}\left[\nabla_{\boldsymbol{\theta}} \log p_{\boldsymbol{\theta}}(\mathbf{x}, \mathbf{z})\right] = \mathbb{E}_{\rho_{\boldsymbol{\theta}}(\mathbf{x},\mathbf{z})}\left[-\frac{\nabla_{\boldsymbol{\theta}}\|\mathbf{x} - \alpha g_{\boldsymbol{\theta}}(\mathbf{z})\|_2^2}{2\sigma^2}\right]. \tag{6}$$

We abbreviate our method as EMD hereafter. To successfully learn the student network with EMD, we need to identify efficient approaches to sample from $\rho_{\boldsymbol{\theta}}(\mathbf{x}, \mathbf{z})$.

### 3.2 Reparametrized sampling and noise cancellation

As an initial strategy, we consider Langevin dynamics which only requires the score functions:

$$\nabla_{\mathbf{x}} \log \rho_{\boldsymbol{\theta}}(\mathbf{x}, \mathbf{z}) = \underbrace{\nabla_{\mathbf{x}} \log q(\mathbf{x})}_{\text{teacher score}} - \underbrace{\nabla_{\mathbf{x}} \log p_{\boldsymbol{\theta}}(\mathbf{x})}_{\text{learned } s_\phi(\mathbf{x})} + \underbrace{\nabla_{\mathbf{x}} \log p_{\boldsymbol{\theta}}(\mathbf{x}|\mathbf{z})}_{-\frac{\mathbf{x} - \alpha g_{\boldsymbol{\theta}}(\mathbf{z})}{\sigma^2}},$$

$$\nabla_{\mathbf{z}} \log \rho_{\boldsymbol{\theta}}(\mathbf{x}, \mathbf{z}) = \nabla_{\mathbf{z}} \log p_{\boldsymbol{\theta}}(\mathbf{x}|\mathbf{z}) + \nabla_{\mathbf{z}} \log p_{\boldsymbol{\theta}}(\mathbf{z}) = -\frac{\mathbf{x} - \alpha g_{\boldsymbol{\theta}}(\mathbf{z})}{\sigma^2}\alpha\nabla_{\mathbf{z}}g_{\boldsymbol{\theta}}(\mathbf{z}) - \mathbf{z}. \tag{7}$$

While we do not have access to the score of the student, $\nabla_{\mathbf{x}} \log p_{\boldsymbol{\theta}}(\mathbf{x})$, we can approximate it with a learned score network $s_\phi$ estimated with denoising score matching as in VSD [26] and Diff-Instruct [22]. As will be covered in Section 3.3, this score network is estimated at all noise levels. The Langevin dynamics defined in Eq. (7) can therefore be simulated at any noise level.

Running Langevin MCMC is expensive and requires careful tuning, and we found this challenging in the context of diffusion model distillation where different noise levels have different optimal step sizes. We leverage a reparametrization of $\mathbf{x}$ and $\mathbf{z}$ to accelerate the joint MCMC sampling and simplify step size tuning, similar to Nijkamp et al. [36], Xiao et al. [42]. Specifically, the parametrization $\mathbf{x} = \alpha g_{\boldsymbol{\theta}}(\mathbf{z}) + \sigma\boldsymbol{\epsilon}$ defines a deterministic transformation from the pair of $(\boldsymbol{\epsilon}, \mathbf{z})$ to the pair of $(\mathbf{x}, \mathbf{z})$, which enables us to push back the joint distribution $\rho_{\boldsymbol{\theta}}(\mathbf{x}, \mathbf{z})$ to the $(\boldsymbol{\epsilon}, \mathbf{z})$-space. The reparameterized distribution is

$$\rho_{\boldsymbol{\theta}}(\boldsymbol{\epsilon}, \mathbf{z}) = \frac{q(\alpha g_{\boldsymbol{\theta}}(\mathbf{z}) + \sigma\boldsymbol{\epsilon})}{p_{\boldsymbol{\theta}}(\alpha g_{\boldsymbol{\theta}}(\mathbf{z}) + \sigma\boldsymbol{\epsilon})}p(\boldsymbol{\epsilon})p(\mathbf{z}). \tag{8}$$

The score functions become

$$\nabla_{\boldsymbol{\epsilon}} \log \rho(\boldsymbol{\epsilon}, \mathbf{z}) = \sigma(\nabla_{\mathbf{x}} \log q(\mathbf{x}) - \nabla_{\mathbf{x}} \log p_{\boldsymbol{\theta}}(\mathbf{x})) - \boldsymbol{\epsilon},$$

$$\nabla_{\mathbf{z}} \log \rho(\boldsymbol{\epsilon}, \mathbf{z}) = \alpha(\nabla_{\mathbf{x}} \log q(\mathbf{x}) - \nabla_{\mathbf{x}} \log p_{\boldsymbol{\theta}}(\mathbf{x}))\nabla_{\mathbf{z}}g_{\boldsymbol{\theta}}(\mathbf{z}) - \mathbf{z}. \tag{9}$$

---

**Algorithm 1:** EM Distillation

---

**Input:** Teacher score functions $\nabla_{\mathbf{x}_t} \log q_t(\mathbf{x}_t)$, generator network $g_{\boldsymbol{\theta}}$, prior $p(\mathbf{z})$, score network $s_\phi$, noise scheduler $p(t)$, weighting functions $w(t)$ and $\tilde{w}(t)$, # of MCMC steps $K$, MCMC step size $\gamma$.

**Output:** Generator network $g_{\boldsymbol{\theta}}$, score network $s_\phi$.

**while** not converged **do**

    Sampling a batch of $t, \mathbf{z}, \boldsymbol{\epsilon}$ from $p(t), p(\mathbf{z}), \mathcal{N}(\mathbf{0}, \boldsymbol{I})$ to obtain $\mathbf{x}_t$

    Updating $s_\phi$ via Stochastic Gradient Descent with the batch estimate of Eq. (12)

    Sampling $\mathbf{x}_t^K$ and $\mathbf{z}^K$ with $(\boldsymbol{\epsilon}, \mathbf{z})$-corrector$(\mathbf{x}_0, \boldsymbol{\epsilon}, \mathbf{z}, t, \nabla_{\mathbf{x}_t} \log q_t(\mathbf{x}_t), g_{\boldsymbol{\theta}}, s_\phi, K, \gamma)$

    Updating $g_{\boldsymbol{\theta}}$ via Stochastic Gradient Ascent with the batch estimate of Eq. (11)

**end while**

---

**Algorithm 2:** $(\boldsymbol{\epsilon}, \mathbf{z})$-corrector

---

**Input:** $\mathbf{x}_0, \boldsymbol{\epsilon}, \mathbf{z}, t$, teacher score function $\nabla_{\mathbf{x}_t} \log q_t(\mathbf{x}_t)$, generator network $g_{\boldsymbol{\theta}}$, prior $p_0(\mathbf{z})$, score network $s_{\boldsymbol{\phi}}$, # of MCMC steps $K$, MCMC step size $\gamma$.
**Output:** $\mathbf{x}_t^K, \mathbf{z}^K$.
Sampling Langevin noise $\mathbf{n}^1, \mathbf{n}^2, ..., \mathbf{n}^K$ from $\mathcal{N}(\mathbf{0}, \boldsymbol{I})$, letting $\boldsymbol{\epsilon}^0 = \boldsymbol{\epsilon}, \mathbf{z}^0 = \mathbf{z}$
**for** $i$ in $[1, K]$ **do**
    Updating $(\boldsymbol{\epsilon}^i, \mathbf{z}^i)$ with 1-step Langevin update over scores Eq. (9), with $\boldsymbol{\epsilon}^i$ updated using $\mathbf{n}^i$
**end for**
Pushing $(\boldsymbol{\epsilon}^K, \mathbf{z}^K)$ forward to $(\mathbf{x}_t^K, \mathbf{z}^K)$ and then canceling the noises in $\mathbf{x}_t^K$

---

See Appendix B for a detailed derivation. We found that this parameterization admits the same step sizes across noise levels and results in better performance empirically (Table 1).

Still, learning the student with these samples continued to present challenges. When visualizing samples $\mathbf{x}$ produced by MCMC (see Fig. 2a), we found that samples contained substantial noise. While this makes sense given the level of noise in the marginal distributions, we found that this inhibited learning of the student. We identify that, due to the structure of Langevin dynamics, there is noise added to $\mathbf{x}$ at each step that can be linearly accumulated across iterations. By removing this accumulated noise along with the temporally decayed initial $\boldsymbol{\epsilon}$, we recover cleaner $\mathbf{x}$ samples (Fig. 2b). Since $\mathbf{x}$ is effectively a regression target in Eq. (6), and

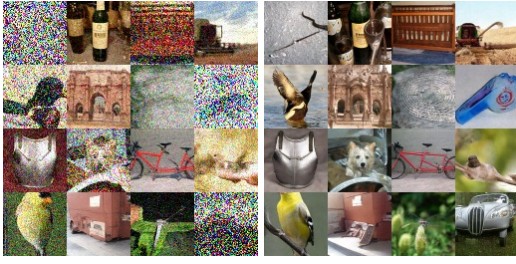

(a) $\mathbf{x}$ w/ accumulated noise    (b) $\mathbf{x}$ w/o accumulated noise
Figure 2: Images after 8-step Langevin updates with and without accumulated noise.

the expected value of the noises is $\mathbf{0}$, canceling these noises reduces variance of the gradient without introducing bias. Empirically, we find bookkeeping the sampled noises in the MCMC chain and canceling these noises after the loop significantly stabilize the training of the generator network. This noise cancellation was critical to the success of EMD, and is detailed in Appendix B and ablated in experiments (Fig. 3ab).

### 3.3 Maximum Likelihood across all noise levels

The derivation above assumes smoothing the data distribution with a single noise level. In practice, the diffusion teachers always employ multiple noise levels $t$, coordinated by a noise schedule $p(t)$. Therefore, we optimize a weighted loss over all noise levels of the diffusion model, to encourage that the marginals of the student network match the marginals of the diffusion process at all noise levels:

$$\nabla_{\boldsymbol{\theta}} \mathcal{L}(\boldsymbol{\theta}) = \nabla_{\boldsymbol{\theta}} \mathbb{E}_{p(t), q_t(\mathbf{x}_t)} [\tilde{w}(t) \log p_{\boldsymbol{\theta}, t}(\mathbf{x}_t)] = \mathbb{E}_{p(t), \rho_t(\mathbf{x}_t, \mathbf{z})} [\tilde{w}(t) \nabla_{\boldsymbol{\theta}} \log p_{\boldsymbol{\theta}, t}(\mathbf{x}_t, \mathbf{z})], \quad (10)$$

where $p_{\boldsymbol{\theta}, t}(\mathbf{x}_t, \mathbf{z})$ are a series of latent-variable models as defined in Section 2.4, with a shared generator $g_{\boldsymbol{\theta}}(\mathbf{z})$ across all noise levels. Empirically, we find $\tilde{w}(t) = \sigma_t^2/\alpha_t$ or $\tilde{w}(t) = \sigma_t^2/\alpha_t^2$ perform well.

Denote the resulted distribution after $K$ steps of MCMC sampling with noise cancellation as $\rho_t^K(\mathbf{x}_t^K, \mathbf{z}^K)$, the final gradient for the generator network $g_{\boldsymbol{\theta}}$ is

$$\nabla_{\boldsymbol{\theta}} \mathcal{L}(\boldsymbol{\theta}) = \mathbb{E}_{p(t), \rho_t^K(\mathbf{x}_t^K, \mathbf{z}^K)} \left[ -\tilde{w}(t) \frac{\nabla_{\boldsymbol{\theta}} \|\mathbf{x}_t^K - \alpha_t g_{\boldsymbol{\theta}}(\mathbf{z}^K)\|_2^2}{2\sigma_t^2} \right]. \quad (11)$$

The final gradient for the score network $s_{\boldsymbol{\phi}}(\mathbf{x}_t, t)$ is

$$\nabla_{\boldsymbol{\phi}} \mathcal{J}(\boldsymbol{\phi}) = \mathbb{E}_{p(t), p_{\boldsymbol{\theta}, t}(\mathbf{x}_t, \mathbf{z})} \left[ w(t) \nabla_{\boldsymbol{\phi}} \|s_{\boldsymbol{\phi}}(\mathbf{x}_t, t) - \nabla_{\mathbf{x}_t} \log p_t(\mathbf{x}_t | g_{\boldsymbol{\theta}}(\mathbf{z}))\|_2^2 \right]. \quad (12)$$

Similar to VSD [26, 22], we employ alternating update for the generator network $g_{\boldsymbol{\theta}}$ and the score network $s_{\boldsymbol{\phi}}(\mathbf{x}_t, t)$. See summarization in Algorithm 1.

## 3.4 Connection with VSD and Diff-Instruct

In this subsection, we reveal an interesting connection between EMD and Variational Score Distillation (VSD) [26, 22], *i.e.*, although motivated by optimizing different types of divergences, VSD [26, 22] is equivalent to EMD with a special sampling scheme.

To see this, consider the 1-step EMD with noise cancellation, stepsize $\gamma = 1$ in $\mathbf{x}$, and no update on $\mathbf{z}$

$$\mathbf{x}_t^0 = \alpha_t g_{\boldsymbol{\theta}}(\mathbf{z}) + \sigma_t \epsilon, \quad \mathbf{x}_t^1 = \alpha_t g_{\boldsymbol{\theta}}(\mathbf{z}) + \sigma_t^2 \nabla_{\mathbf{x}} \log \frac{q(\mathbf{x}_t^0)}{p_{\boldsymbol{\theta},t}(\mathbf{x}_t^0)} + \sqrt{2}\sigma \mathbf{n}^\top. \tag{13}$$

Substitute it into Eq. (11), we have

$$\begin{aligned}
\nabla_{\boldsymbol{\theta}} \mathcal{L}(\boldsymbol{\theta}) &= \mathbb{E}_{p(t),p(\epsilon),p(\mathbf{z})} \left[ -\tilde{w}(t) \frac{\nabla_{\boldsymbol{\theta}} \|\mathbf{x}_t^1 - \alpha_t g_{\boldsymbol{\theta}}(\mathbf{z})\|_2^2}{2\sigma_t^2} \right] \\
&= \mathbb{E}_{p(t),p(\epsilon),p(\mathbf{z})} \left[ \tilde{w}(t)(\nabla_{\mathbf{x}_t} \log q_t(\mathbf{x}_t) - \nabla_{\mathbf{x}_t} \log p_{\boldsymbol{\theta},t}(\mathbf{x}_t))\alpha_t \nabla_{\boldsymbol{\theta}} g_{\boldsymbol{\theta}}(\mathbf{z}) \right],
\end{aligned} \tag{14}$$

which is exactly the gradient for VSD (Eq. (5)), up to a sign difference. This insight demonstrates that, EMD framework can flexibly interpolate between mode-seeking and mode-covering divergences, by leveraging different sampling schemes from 1-step sampling in only $\mathbf{x}$ (a likely biased sampler) to many-step joint sampling in $(\mathbf{x}, \mathbf{z})$ (closer to a mixing sampler). Notably, for image generation, some believe that *forward* KL divergence may fail to achieve better fidelity compared to *reverse* KL divergence. The interpolation enabled by EMD can thus be very useful in practice.

If we further assume the marginal $p_{\boldsymbol{\theta}}(\mathbf{x})$ is a Gaussian, then EMD update in Eq. 14 would resemble Score Distillation Sampling (SDS) [9].

## 4 Related Work

**Diffusion acceleration.** Diffusion models have the notable issue of slowness in inference, which motivates many research efforts to accelerate the sampling process. One line of work focuses on developing numerical solvers [43, 12, 44–47] for the PF-ODE. Another line of work leverages the concept of knowledge distillation [48] to condense the sampling trajectory of PF-ODE into fewer steps [13, 49, 15, 14, 50, 51, 18, 52–55]. However, both approaches have significant limitations and have difficulty in substantially reducing the sampling steps to the single-step regime without significant loss in perceptual quality.

**Single-step diffusion models.** Recently, several methods for one-step diffusion sampling have been proposed, sharing the same goal as our approach. Some methods fine-tune the pre-trained diffusion model into a single-step generator via adversarial training [20, 21, 56], where the adversarial loss enhances the sharpness of the diffusion model's single-step output. Adversarial training can also be combined with trajectory distillation techniques to improve performance in few or single-step regimes [52, 57, 58]. Score distillation techniques [9, 26] have been adopted to match the distribution of the one-step generator's output with that of the teacher diffusion model, enabling single-step generation [22, 41]. Additionally, Yin et al. [23] introduces a regression loss to further enhance performance. These methods achieve more impressive 1-step generation, some of which enjoy additional merits of being data-free or flexible in the selection of generator architecture. However, they often minimizes over mode-seeking divergences that can fail to capture the full distribution and therefore causes mode collapse issues. We discuss the connection between our method and this line of work in Section 3.4. Concurrent with our work, Zhou et al. [59] adopt Fisher divergence as the distillation objective and propose a novel decomposition that alleviates the dependency on the approximation accuracy of the auxiliary score network. Although the adopted Fisher divergence is similar to *reverse* KL in terms of the reparametrization and hence the risk of mode collapse, Zhou et al. [59] demonstrate impressive performance gain.

## 5 Experiments

We employ EMD to learn one-step image generators on ImageNet 64×64, ImageNet 128×128 [60] and text-to-image generation. The student generators are initialized with the teacher model weights. Results are compared according to Frechet Inception Distance (FID) [61], Inception Score (IS) [62],

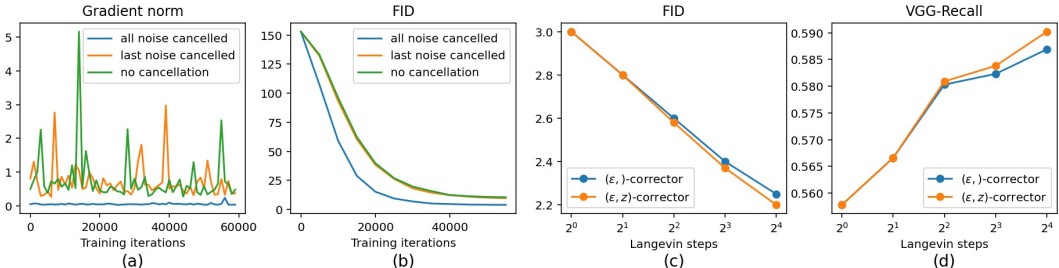

Figure 3: (a)(b) Gradient norms and FIDs for complete noise cancellation, last-step noise cancellation and no noise cancellation. (c)(d) FIDs and Recalls of EMD with different numbers of Langevin steps.

Recall (Rec.) [63] and CLIP Score [64]. Throughout this section, we will refer to the proposed EMD with $K$ steps of Langevin updates on $(\mathbf{x}, \mathbf{z})$ as EMD-$K$, and we use EMD-1 to describe the DiffInstruct/VSD-equivalent formulation with only one update in $\mathbf{x}$ as presented in Section 3.4.

## 5.1 ImageNet

We start from showcasing the effect of the key components of EMD, namely noise cancellation, multi-step joint sampling, and reparametrized sampling. We then summarize results on ImageNet 64×64 with Karras et al. [47] as teacher, and ImageNet 128×128 with Kingma and Gao [29] as teacher.

**Noise cancellation**  During our development, we observed the vital importance of canceling the noise after the Langevin update. Even though theoretically speaking our noise cancellation technique does not guarantee reducing the variance of the gradients for learning, we find removing the accumulated noise term from the samples (including the initial diffusion noise $\boldsymbol{\epsilon}$) does give us seemingly clean images empirically. See Fig. 2 for a comparison. These updated $\mathbf{x}^K$ can be seen as regression targets in Eq. (11). Intuitively speaking, regressing a generator towards clean images should result in more stable training than towards noisy images. Reflected in the training process, canceling the noise significantly decreases the variance in the gradient (Fig. 3a) and boosts the speed of convergence (Fig. 3b). We also compare with another setting where only the noise in the last step gets canceled, which is only marginally helpful.

**Multi-step joint sampling**  We scrutinize the effect of multi-step joint update on $(\boldsymbol{\epsilon}, \mathbf{z})$. Empirically, we find a constant step size of Langevin dynamics across all noise levels in the $(\boldsymbol{\epsilon}, \mathbf{z})$-space works well: $\boldsymbol{\gamma} = (\gamma_{\boldsymbol{\epsilon}}, \gamma_{\mathbf{z}}) = (0.4^2, 0.004^2)$, which simplifies the process of step size tuning. Fig. 1 shows results of running this $(\boldsymbol{\epsilon}, \mathbf{z})$-corrector for 300 steps. We can see that the $(\boldsymbol{\epsilon}, \mathbf{z})$-corrector removes visual artifacts and improves structure. Fig. 3cd illustrates the relation between the distilled generator's performance and the number of Langevin steps per distillation iteration, measured by FID and Recall respectively. Both metrics show clear improvement monotonically as the number of Langevin steps increases. Recall is designed for measuring mode coverage [63], and has been widely adopted in the GAN literature. A larger number of Langevin steps encourages better mode coverage, likely because it approximates the mode-covering *forward* KL better. Sampling $\mathbf{z}$ is more expensive than sampling $\boldsymbol{\epsilon}$, requiring back-propagation through the generator $g_{\boldsymbol{\theta}}$. An alternative is to only sample $\boldsymbol{\epsilon}$ while keeping $\mathbf{z}$ fixed, with the hope that if $\mathbf{x}$ does not change dramatically with a finite number of MCMC updates, the initial $\mathbf{z}$ remains a good approximation of samples from $\rho_{\boldsymbol{\theta}}(\mathbf{z}|\mathbf{x})$. As shown in Fig. 3cd, sampling $\boldsymbol{\epsilon}$ performs similarly to the joint sampling of $(\boldsymbol{\epsilon}, \mathbf{z})$ when the number of sampling steps is small, but starts to fall behind with more sampling steps.

**Reparametrized sampling**  As shown in Appendix B, the noise cancellation technique does not depend on the reparametrization. One can start from either the score functions of $(\mathbf{x}, \mathbf{z})$ in Eq. (7) or the score functions of $(\boldsymbol{\epsilon}, \mathbf{z})$ in Eq. (9) to derive something similar. We conduct a comparison between the two parameterizations for joint sampling, $(\mathbf{x}, \mathbf{z})$-corrector and $(\boldsymbol{\epsilon}, \mathbf{z})$-corrector.

For the $(\mathbf{x}, \mathbf{z})$-corrector, we set the step size of $\mathbf{x}$ as $\sigma_t^2 \gamma_{\boldsymbol{\epsilon}}$ to align the magnitude of update with the one of the $(\boldsymbol{\epsilon}, \mathbf{z})$-corrector, and keep

Table 1: EMD-8 on ImageNet 64×64, 100k steps of training

|  | FID ($\downarrow$) | IS ($\uparrow$) |
|---|---|---|
| $(\mathbf{x}, )/(\boldsymbol{\epsilon}, )$ | 2.829 | 62.31 |
| $(\mathbf{x}, \mathbf{z})$ | 3.11 | 61.08 |
| $(\boldsymbol{\epsilon}, \mathbf{z})$ | **2.77** | **62.98** |

Table 2: Class-conditional genreation on ImageNet 64×64.

| Method | NFE (↓) | FID (↓) | Rec. (↑) |
|---|---|---|---|
| *Multiple Steps* | | | |
| DDIM [12] | 50 | 13.7 | |
| EDM-Heun [47] | 10 | 17.25 | |
| DPM Solver [44] | 10 | 7.93 | |
| PD [13] | 2 | 8.95 | 0.65 |
| CD [15] | 2 | 4.70 | 0.64 |
| Multistep CD [18] | 2 | 2.0 | - |
| *Single Step* | | | |
| BigGAN-deep [65] | 1 | 4.06 | 0.48 |
| EDM [47] | 1 | 154.78 | - |
| PD [13] | 1 | 15.39 | 0.62 |
| BOOT [16] | 1 | 16.30 | 0.36 |
| DFNO [17] | 1 | 7.83 | - |
| TRACT [14] | 1 | 7.43 | - |
| CD-LPIPS [15] | 1 | 6.20 | 0.63 |
| Diff-Instruct [22] | 1 | 5.57 | - |
| DMD [23] | 1 | 2.62 | - |
| EMD-1 (baseline) | 1 | 3.1 | 0.55 |
| **EMD-16 (ours)** | 1 | **2.20** | 0.59 |
| Teacher | 256 | 1.43 | - |

Table 4: FID-30k for text-to-image generation in MSCOCO. [†] Results are evaluated by Yin et al. [23].

| Family | Method | Latency (↓) | FID (↓) |
|---|---|---|---|
| Unaccelerated | DALL·E [66] | - | 27.5 |
| | DALL·E 2 [4] | - | 10.39 |
| | Parti-3B [67] | 6.4s | 8.10 |
| | Make-A-Scene [68] | 25.0s | 11.84 |
| | GLIDE [69] | 15.0s | 12.24 |
| | Imagen [5] | 9.1s | 7.27 |
| | eDiff-I [70] | 32.0s | 6.95 |
| GANs | StyleGAN-T [71] | 0.10s | 13.90 |
| | GigaGAN [72] | 0.13s | 9.09 |
| Accelerated | DPM++ (4 step)[†] [45] | 0.26s | 22.36 |
| | UniPC (4 step)[†] [73] | 0.26s | 19.57 |
| | LCM-LoRA (1 step)[†] [74] | 0.09s | 77.90 |
| | LCM-LoRA (4 step)[†] [74] | 0.19s | 23.62 |
| | InstaFlow-0.9B[†] [55] | 0.09s | 13.10 |
| | UFOGen [20] | 0.09s | 12.78 |
| | DMD (tCFG=3)[†] [23] | 0.09s | 11.49 |
| | EMD-1 (baseline, tCFG=3) | 0.09s | 10.96 |
| | EMD-1 (baseline, tCFG=2) | 0.09s | 9.78 |
| | **EMD-8 (ours, tCFG=2)** | 0.09s | **9.66** |
| Teacher | SDv1.5[†] [6] | 2.59s | 8.78 |

Table 3: Class-conditional generation on ImageNet 128×128.

| Method | NFE (↓) | FID (↓) | IS (↑) |
|---|---|---|---|
| *Multiple Steps* | | | |
| Multistep CD [18] | 8 | 2.1 | 164 |
| Multistep CD [18] | 4 | 2.3 | 157 |
| Multistep CD [18] | 2 | 3.1 | 147 |
| *Single Step* | | | |
| CD [15] | 1 | 7.0 | - |
| EMD-1 (baseline) | 1 | 6.3 | $134 \pm 2.75$ |
| **EMD-16 (ours)** | 1 | **6.0** | **$140 \pm 2.83$** |
| Teacher | 512 | 1.75 | $171.1 \pm 2.7$ |

Table 5: CLIP Score in high CFG regime.

| Family | Method | Latency (↓) | CLIP (↑) |
|---|---|---|---|
| Accelerated | DPM++ (4 step) [45][†] | 0.26s | 0.309 |
| | UniPC (4 step)[†] [73] | 0.26s | 0.308 |
| | LCM-LoRA (1 step)[†] [74] | 0.09s | 0.238 |
| | LCM-LoRA (4 step)[†] [74] | 0.19s | 0.297 |
| | DMD[†] [23] | 0.09s | 0.320 |
| | **EMD-8 (ours)** | 0.09s | 0.316 |
| Teacher | SDv1.5 [†] [6] | 2.59s | 0.322 |

the step size of $\mathbf{z}$ the same (see Appendix B for details). This also promotes numerical stability in $(\mathbf{x}, \mathbf{z})$-corrector by canceling the $\sigma_t^2$ in the denominator of the term $\nabla_{\mathbf{x}} \log p_{\boldsymbol{\theta}}(\mathbf{x}|\mathbf{z}) = -\frac{\mathbf{x} - \alpha g_{\boldsymbol{\theta}}(\mathbf{z})}{\sigma^2}$ in the score function (Eq. (7)). A similar design choice was proposed in Song and Ermon [27]. Also note that adjusting the step sizes in this way results in an equivalence between $(\boldsymbol{\epsilon}, )$-corrector and $(\mathbf{x}, )$-corrector without sampling in $\mathbf{z}$, which serves as the baseline for the joint sampling.

Table 1 reports the quantitative comparisons with EMD-8 on ImageNet 64×64 after 100k steps of training. While joint sampling with $(\boldsymbol{\epsilon}, \mathbf{z})$-corrector improves over $(\boldsymbol{\epsilon}, )$-corrector, $(\mathbf{x}, \mathbf{z})$-corrector struggles to even match the baseline. Possible explanations include that the space of $(\boldsymbol{\epsilon}, \mathbf{z})$ is more MCMC friendly, or it requires more effort on searching for the optimal step size of $\mathbf{z}$ for the $(\mathbf{x}, \mathbf{z})$-corrector. We leave further investigation to future work.

**Comparsion with existing methods** We report the results from our full-fledged method, EMD-16, which utilizes a $(\boldsymbol{\epsilon}, \mathbf{z})$-corrector with 16 steps of Langevin updates, and compare with existing approaches. We train for 300k steps on ImageNet 64×64, and 200k steps on ImageNet 128×128. Other hyperparameters can be found in Appendix C. Samples from the distilled generator can be found in Fig. 4. We also include additional samples in Appendix D.1. We summarize the comparison with existing methods for few-step diffusion generation in Table 2 and Table 3 for ImageNet 64×64 and ImageNet 128×128 respectively. Note that we also tune the baseline EMD-1, which in formulation is equivalent to Diff-Instruct [22], to perform better than their reported numbers. The improvement mainly comes from a fine-grained tuning of learning rates and enabling dropout for both the teacher and student score functions. Our final models outperform existing approaches for one-step distillation of diffusion models in terms of FID scores on both tasks. On ImageNet $64 \times 64$, EMD achieves a competitive recall among distribution matching approaches but falls behind trajectory distillation approaches which maintain individual trajectory mappings from the teacher.

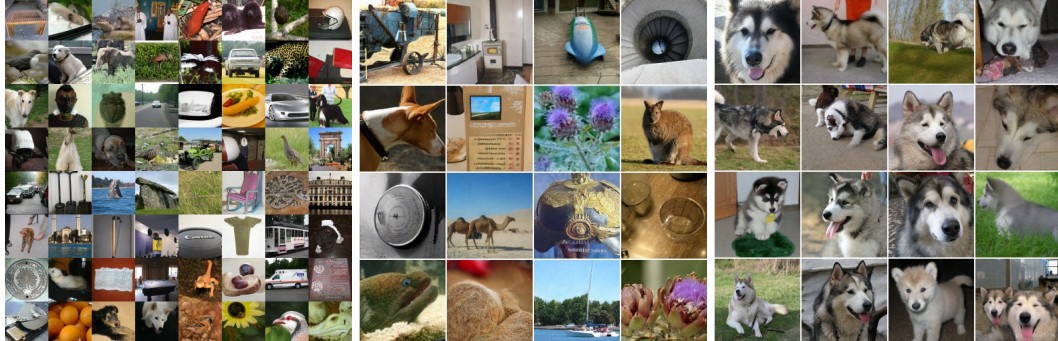

(a) ImageNet 64×64 multi-class     (b) ImageNet 128×128 multi-class     (c) ImageNet 128×128 single-class

Figure 4: ImageNet samples from the distilled 1-step generator. Models are trained class-conditionally with all classes. We provide single-class samples in (c) to demonstrate good mode coverage.

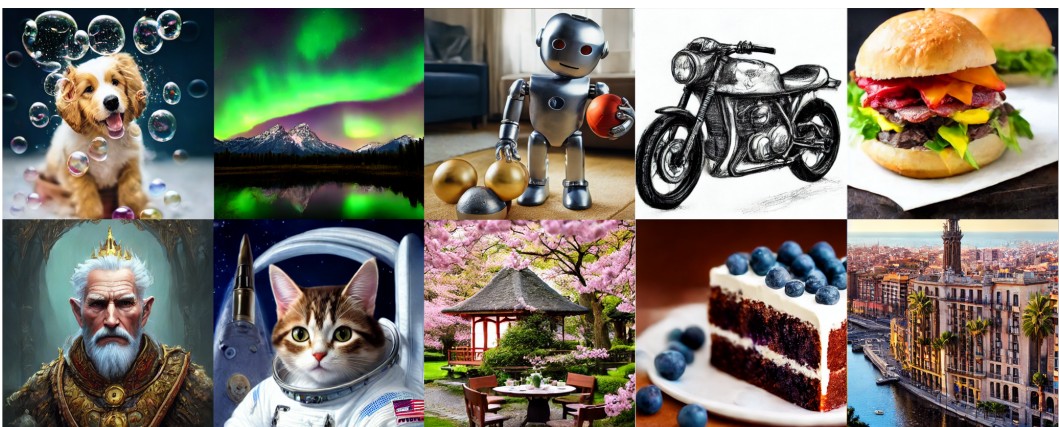

Figure 5: Text-to-image samples from the 1-step student model distilled from Stable Diffusion v1.5.

## 5.2 Text-to-image generation

We further test the potential of EMD on text-to-image models at scale by distilling the Stable Diffusion v1.5 [6] model. Note that the training is image-free and we only use text prompts from the LAION-Aesthetics-6.25+ dataset [75]. On this task, DMD [23] is a strong baseline, which introduced an additional regression loss to VSD or Diff-Instruct to avoid mode collapse. However, we find the baseline without regression loss, or equivalently EMD-1, can be improved by simply tuning the hyperparameter $t^*$. Empirically, we find it is better to set $t^*$ to intermediate noise levels, consistent with the observation from Luo et al. [22]. In Appendix C.4 we discuss the selection of $t^*$. The intuition is that by choosing the value of $t^*$, we choose a specific denoiser at that noise level for initialization. Other hyperparameters can be found in Appendix C.3.

We evaluate the distilled one-step generator for text-to-image generation with zero-shot generalization on MSCOCO [76] and report the FID-30k in Table 4 and CLIP Score in Table 5. Yin et al. [23] uses the guidance scale of 3.0 to compose the classifer-free guided teacher score (we refer to this guidance scale of teacher as tCFG) in the learning gradient of DMD, for it achieves the best FID for DDIM sampler. However, we find EMD achieves a lower FID at the tCFG of 2.0. Our method, EMD-8, trained on 256 TPU-v5e for 5 hours (5000 steps), achieves the FID=9.66 for one-step text-to-image generation. Using a higher tCFG, similar to DMD, produces a model with competitive CLIP Score. In Fig. 5, we include some samples for qualitative evaluation. Additional qualitative results (Tables 14 and 15), as well as side-by-side comparisons (Tables 10 to 13) with trajectory-based distillation baselines [55, 74] and adversarial distillation baselines [21] can be found in Appendix D.2.

Table 6: Training steps per second in ablations for computation overhead in ImageNet 64×64

| Algorithmic Ablation | sec/step |
|---|---|
| Student score matching only | 0.303 |
| Generator update for EMD-1 based on $(\epsilon, \mathbf{z})$-corrector | 0.303 |
| Generator update for EMD-2 based on $(\epsilon, \mathbf{z})$-corrector | 0.417 |
| Generator update for EMD-4 based on $(\epsilon, \mathbf{z})$-corrector | 0.556 |
| Generator update for EMD-8 based on $(\epsilon, \mathbf{z})$-corrector | 0.714 |
| Generator update for EMD-16 based on $(\epsilon, \mathbf{z})$-corrector | 1.111 |
| EMD-16 ( student score matching + generator update based on $(\epsilon, \mathbf{z})$-corrector) | 1.515 |
| Baseline Diff-Instruct (student score matching + generator update) | 0.703 |

## 5.3 Computation overhead in training

Despite EMD being more expensive per training iteration compared to the baseline approach Diff-Instruct, we find the performance gain of EMD cannot be realized by simply running Diff-Instruct for the same amount of time or even longer than EMD. In fact, the additional computational cost that EMD introduced is moderate even with the most expensive EMD-16 setting. In Table 6 we report some quantitative measurement of the computation overhead. Since it is challenging to time each python method's wall-clock time in our infrastructure, we instead logged the sec/step for experiments with various algorithmic ablations on ImageNet 64×64. EMD-16 only doubles the wall-clock time of Diff-Instruct when taking all other overheads into account.

## 6 Discussion and limitation

We present EMD, a maximum likelihood-based method that leverages EM framework with novel sampling and optimization techniques to learn a one-step student model whose marginal distributions match the marginals of a pretrained diffusion model. EMD demonstrates strong performance in class-conditional generation on ImageNet and text-to-image generation. Despite exhibiting compelling results, EMD has a few limitations that call for future work. Empirically, we find that EMD still requires the student model to be initialized from the teacher model to perform competitively, and is sensitive to the choice of $t^*$ (fixed timestep conditioning that repurposes the diffusion denoiser to become a one-step genertor) at initialization. While training a student model entirely from scratch is supported theoretically by our framework, empirically we were unable to achieve competitive results. Improving methods to enable generation from randomly initialized generator networks with distinct architectures and lower-dimensional latent variables is an exciting direction of future work. Although being efficient in inference, EMD introduces additional computational cost in training by running multiple sampling steps per iteration, and the step size of MCMC sampling can require careful tuning. There remains a fundamental trade-off between training cost and model performance. Analysis and further improving on the Pareto frontier of this trade-off would be interesting for future work.

## Acknowledgement

We thank Jonathan Heek and Lucas Theis for their valuable discussion and feedback. We also thank Tianwei Yin for helpful sharing of experimental details in his work. Sirui would like to thank Tao Zhu, Jiahui Yu, and Zhishuai Zhang for their support in a prior internship at Google DeepMind.

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

## A    Expectation-Maximization

To learn a latent variable model $p_{\boldsymbol{\theta}}(\mathbf{x}, \mathbf{z}) = p_{\boldsymbol{\theta}}(\mathbf{x}|\mathbf{z})p_{\boldsymbol{\theta}}(\mathbf{z})$, $p_{\boldsymbol{\theta}}(\mathbf{x}) = \int p_{\boldsymbol{\theta}}(\mathbf{x}, \mathbf{z})d\mathbf{z}$ from a target distribution $q(\mathbf{x})$, the EM-like transformation on the gradient of the log-likelihood function is:

$$
\begin{aligned}
\nabla_{\boldsymbol{\theta}}\mathcal{L}(\boldsymbol{\theta}) &= \nabla_{\boldsymbol{\theta}}\, \mathbb{E}_{q(\mathbf{x})}[\log p_{\boldsymbol{\theta}}(\mathbf{x})] \\
&= \mathbb{E}_{p_{\boldsymbol{\theta}}(\mathbf{z}|\mathbf{x})}[\mathbb{E}_{q(\mathbf{x})}[\nabla_{\boldsymbol{\theta}}\log p_{\boldsymbol{\theta}}(\mathbf{x})]] \\
&= \mathbb{E}_{q(\mathbf{x})p_{\boldsymbol{\theta}}(\mathbf{z}|\mathbf{x})}[\nabla_{\boldsymbol{\theta}}\log p_{\boldsymbol{\theta}}(\mathbf{x}) + \nabla_{\boldsymbol{\theta}}\log p_{\boldsymbol{\theta}}(\mathbf{z}|\mathbf{x})] \\
&= \mathbb{E}_{q(\mathbf{x})p_{\boldsymbol{\theta}}(\mathbf{z}|\mathbf{x})}[\nabla_{\boldsymbol{\theta}}\log p_{\boldsymbol{\theta}}(\mathbf{x}, \mathbf{z})].
\end{aligned}
\tag{15}
$$

Line 3 is due to the simple equality that $\mathbb{E}_{p_{\boldsymbol{\theta}}(\mathbf{z}|\mathbf{x})}[\nabla_{\boldsymbol{\theta}}\log p_{\boldsymbol{\theta}}(\mathbf{z}|\mathbf{x})] = 0$.

## B    Reparametrized sampling and noise cancellation

**Reparametrization.**    The EM-like algorithm we propose requires joint sampling of $(\mathbf{x}, \mathbf{z})$ from $\rho_{\boldsymbol{\theta}}(\mathbf{x}, \mathbf{z})$. Similar to [36, 42], we utilize a reparameterization of $\mathbf{x}$ and $\mathbf{z}$ to overcome challenges in joint MCMC sampling, such as slow convergence and complex step size tuning. Notice that $\mathbf{x} = \alpha g_{\boldsymbol{\theta}}(\mathbf{z}) + \sigma\boldsymbol{\epsilon}$ defines a deterministc mapping from $(\boldsymbol{\epsilon}, \mathbf{z})$ to $(\mathbf{x}, \mathbf{z})$. Then by *change of variable* we have:

$$
\begin{aligned}
\rho_{\boldsymbol{\theta}}(\boldsymbol{\epsilon}, \mathbf{z})d\boldsymbol{\epsilon}d\mathbf{z} &= \rho_{\boldsymbol{\theta}}(\mathbf{x}, \mathbf{z})d\mathbf{x}d\mathbf{z} \\
&= \frac{q(\mathbf{x})}{p_{\boldsymbol{\theta}}(\mathbf{x})}p_{\boldsymbol{\theta}}(\mathbf{x}, \mathbf{z})d\mathbf{x}d\mathbf{z} \\
&= \frac{q(\alpha g_{\boldsymbol{\theta}}(\mathbf{z}) + \sigma\boldsymbol{\epsilon})}{p_{\boldsymbol{\theta}}(\alpha g_{\boldsymbol{\theta}}(\mathbf{z}) + \sigma\boldsymbol{\epsilon})}p_{\boldsymbol{\theta}}(\boldsymbol{\epsilon}, \mathbf{z})d\boldsymbol{\epsilon}d\mathbf{z} \\
\Rightarrow \rho_{\boldsymbol{\theta}}(\boldsymbol{\epsilon}, \mathbf{z}) &= \frac{q(\alpha g_{\boldsymbol{\theta}}(\mathbf{z}) + \sigma\boldsymbol{\epsilon})}{p_{\boldsymbol{\theta}}(\alpha g_{\boldsymbol{\theta}}(\mathbf{z}) + \sigma\boldsymbol{\epsilon})}p(\boldsymbol{\epsilon})p(\mathbf{z}),
\end{aligned}
\tag{16}
$$

where $p(\boldsymbol{\epsilon})$ and $p(\mathbf{z})$ are standard Normal distributions.

The score functions become

$$\nabla_{\boldsymbol{\epsilon}} \log \rho(\boldsymbol{\epsilon}, \mathbf{z}) = \sigma(\nabla_{\mathbf{x}} \log q(\mathbf{x}) - \nabla_{\mathbf{x}} \log p_{\boldsymbol{\theta}}(\mathbf{x})) - \boldsymbol{\epsilon},$$
$$\nabla_{\mathbf{z}} \log \rho(\boldsymbol{\epsilon}, \mathbf{z}) = \alpha(\nabla_{\mathbf{x}} \log q(\mathbf{x}) - \nabla_{\mathbf{x}} \log p_{\boldsymbol{\theta}}(\mathbf{x}))\nabla_{\mathbf{z}} g_{\boldsymbol{\theta}}(\mathbf{z}) - \mathbf{z}. \tag{17}$$

**Noise cancellation.** The single-step Langevin update on $\boldsymbol{\epsilon}$ is then:

$$\boldsymbol{\epsilon}^{i+1} = (1-\gamma)\boldsymbol{\epsilon}^i + \gamma\sigma\nabla_{\mathbf{x}} \log \frac{q(\mathbf{x}^i)}{p_{\boldsymbol{\theta}}(\mathbf{x}^i)} + \sqrt{2\gamma}\mathbf{n}^i. \tag{18}$$

Interestingly, we find the particular form of $\nabla_{\boldsymbol{\epsilon}} \log \rho(\boldsymbol{\epsilon}, \mathbf{z})$ results in a closed-form accumulation of multi-step updates

$$\boldsymbol{\epsilon}^{i+1} = (1-\gamma)^{i+1}\boldsymbol{\epsilon}^0 + \gamma\sum_{k=0}^{i}(1-\gamma)^{i-k}\sigma\nabla_{\mathbf{x}} \log \frac{q(\mathbf{x}^i)}{p_{\boldsymbol{\theta}}(\mathbf{x}^i)} + \sum_{k=0}^{i}(1-\gamma)^{i-k}\sqrt{2\gamma}\mathbf{n}^k, \tag{19}$$

which, after the push-forward, gives us

$$\mathbf{x}^{i+1} = \alpha g(\mathbf{z}^{i+1}) + \underbrace{\gamma\sum_{k=0}^{i}(1-\gamma)^{i-k}\sigma^2\nabla_{\mathbf{x}} \log \frac{q(\mathbf{x}^i)}{p_{\boldsymbol{\theta}}(\mathbf{x}^i)}}_{drift}$$
$$+ \underbrace{(1-\gamma)^{i+1}\sigma\boldsymbol{\epsilon}^0 + \sum_{k=0}^{i}(1-\gamma)^{i-k}\sqrt{2\gamma}\sigma\mathbf{n}^k}_{noise}, \tag{20}$$

As $\mathbf{x}^{i+1}$ is effectively a regression target in Eq. (6), and the expected value of the $noise$ is $\mathbf{0}$, we can remove it without biasing the gradient. Empirically, we find book-keeping the sampled noises in the MCMC chain and canceling these noises after the loop significantly stabilize the training of the generator network.

The same applies to the $(\mathbf{x}, \mathbf{z})$ sampling (with step size $\gamma\sigma^2$):

$$\mathbf{x}^{i+1} = \mathbf{x}^i + \gamma\sigma^2(\nabla_{\mathbf{x}} \log q(\mathbf{x}^i) - \nabla_{\mathbf{x}} \log p_{\boldsymbol{\theta}}(\mathbf{x}^i)) - \gamma(\mathbf{x}^i - \alpha g(\mathbf{z}^i)) + \sqrt{2\gamma}n^i$$
$$= \gamma\sum_{k=1}^{i}(1-\gamma)^{i-k}\alpha g(\mathbf{z}^k) + \gamma\sum_{k=0}^{i}(1-\gamma)^{i-k}\sigma^2(\nabla_{\mathbf{x}} \log q(\mathbf{x}^i) - \nabla_{\mathbf{x}} \log p_{\boldsymbol{\theta}}(\mathbf{x}^i))$$
$$+ (1-\gamma)^i\alpha g(\mathbf{z}^0) + \underbrace{(1-\gamma)^{i+1}\sigma\boldsymbol{\epsilon}^0 + \sum_{k=0}^{i}(1-\gamma)^{i-k}\sqrt{2\gamma}\sigma n^k}_{noises}. \tag{21}$$

## C  Implementation details

### C.1  ImageNet 64×64

We train the teacher model using the best setting of EDM [47] with the ADM UNet architecture [77]. We inherit the noise schedule and the score matching weighting function from the teacher. We run the distillation training for 300k steps (roughly 8 days) on 64 TPU-v4. We use $(\boldsymbol{\epsilon}, \mathbf{z})$-corrector, in which both the teacher and the student score networks have a dropout probability of 0.1. We list other hyperparameters in Table 7. Instead of listing $t^*$, we list the corresponding log signal-to-noise ratio $\lambda^*$.

Table 7: Hyperparameters for EMD on ImageNet 64×64.

| $lr_g$ | $lr_s$ | batch size | Adam $b_1$ | Adam $b_2$ | $\gamma_{\boldsymbol{\epsilon}}$ | $\gamma_{\mathbf{z}}$ | $K$ | $\lambda^*$ | $\tilde{w}(t)$ |
|---|---|---|---|---|---|---|---|---|---|
| $2\times 10^{-6}$ | $1\times 10^{-5}$ | 128 | 0.0 | 0.99 | $0.4^2$ | $0.004^2$ | 16 | $-3.2189$ | $\frac{\sigma_t^2}{\alpha_t^2}$ |

## C.2 ImageNet 128×128

We train the teacher model following the best setting of VDM++ [29] with the 'U-ViT, L' architecture [78]. We use the 'cosine-adjusted' noise schedule [78] and 'EDM monotonic' weighting for student score matching. We run the distillation training for 200k steps (roughly 10 days) on 128 TPU-v5p. We use $(\epsilon, \mathbf{z})$-corrector, in which both the teacher and the student score networks have a dropout probability of 0.1. We list other hyperparameters in Table 8.

Table 8: Hyperparameters for EMD on ImageNet 128×128.

| $lr_g$ | $lr_s$ | batch size | Adam $b_1$ | Adam $b_2$ | $\gamma_\epsilon$ | $\gamma_\mathbf{z}$ | $K$ | $\lambda^*$ | $\tilde{w}(t)$ |
|---|---|---|---|---|---|---|---|---|---|
| $2 \times 10^{-6}$ | $1 \times 10^{-5}$ | 1024 | 0.0 | 0.99 | $0.4^2$ | $0.004^2$ | 16 | $-6$ | $\frac{\sigma_t^2}{\alpha_t}$ |

## C.3 Text-to-image generation

We adopt the public checkpoint of Stable Diffusion v1.5 [6] as the teacher. We inherit the noise schedule from the teacher model. The student score matching uses the same weighting function as the teacher. We list other hyperparameters in Table 9.

Table 9: Hyperparameters for EMD on Text-to-image generation.

| $lr_g$ | $lr_s$ | batch size | Adam $b_1$ | Adam $b_2$ | $\gamma_\epsilon$ | $\gamma_\mathbf{z}$ | $K$ | $t^*$ | $\tilde{w}(t)$ |
|---|---|---|---|---|---|---|---|---|---|
| $2 \times 10^{-6}$ | $1 \times 10^{-5}$ | 1024 | 0.0 | 0.99 | $0.3^2$ | $0.005^2$ | 8 | 500 | $\frac{\sigma_t^2}{\alpha_t}$ |

## C.4 Choosing $t^*$ and $\lambda^*$

The intuition is that by choosing the value of $t^*$, we choose a specific denoiser at that noise level. When parametrizing $t$, the log-signal-to-noise ratio $\lambda$ is more useful when designing noise schedules, a strictly monotonically decreasing function [28]. Due to the monotonicity, $\lambda^*$ is an alternative representation for $t^*$ that actually reflects the noise levels more directly.

Fig. 6 shows the denoiser generation at the 0th training iteration for different $\lambda^*$ in ImageNet 128×128. When $\lambda^* = 0$, the generated images are no different from Gaussian noises. When $\lambda^* = -6$, the generated images have more details than $\lambda^* = -10$. In the context of EMD, these samples help us understand the initialization of MCMC. According to our experiments, setting $\lambda^* \in [-6, -3]$ results in similar performance. For the numbers reported in the manuscript, we used the same $\lambda^*$ as the baseline Diff-Instruct on ImageNet 64×64 and only did a very rough grid search on ImageNet 128×128 and Text-to-image.

# D Additional qualitative results

## D.1 Additional ImageNet results

In this section, we present additional qualitative samples for our one-step generator on ImageNet 64×64 and ImageNet 128×128 in Fig. 7 to help further evaluate the generation quality and diversity.

## D.2 Additional text-to-image results

In this section, we present additional qualitative samples from our one-step generator distilled from Stable Diffusion 1.5. In Table 10, 11, 12, and 13, we visually compare the sample quality of our method with open-source competing methods for few- or single-step generation. We also include the teacher model in our comparison. We use the public checkpoints of LCM[1] and InstaFlow[2], where

---

[1] https://huggingface.co/latent-consistency/lcm-lora-sdv1-5
[2] https://huggingface.co/XCLiu/instaflow_0_9B_from_sd_1_5

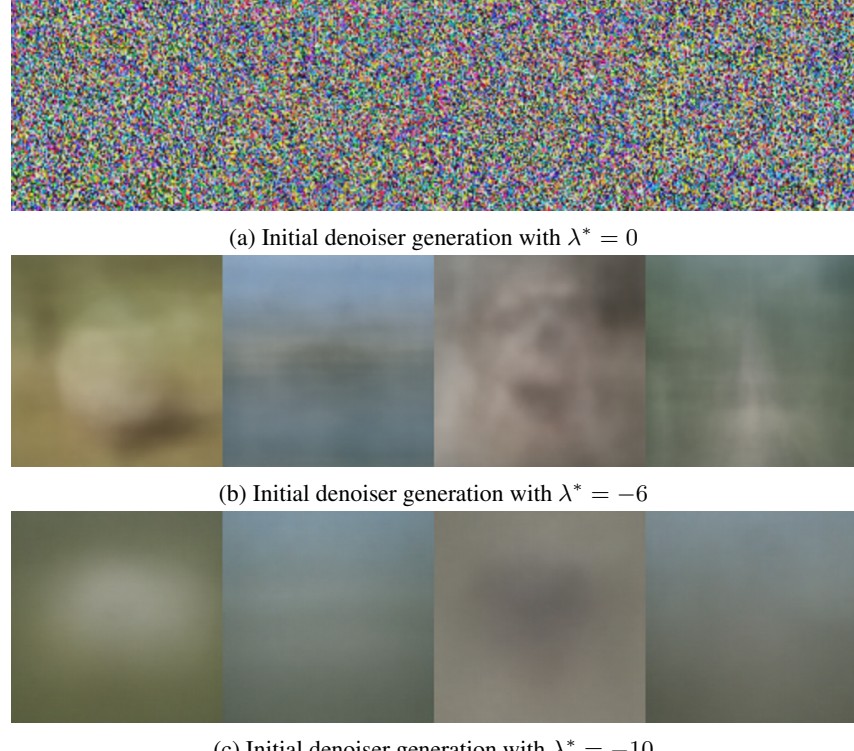

(a) Initial denoiser generation with $\lambda^* = 0$

(b) Initial denoiser generation with $\lambda^* = -6$

(c) Initial denoiser generation with $\lambda^* = -10$

Figure 6: Initial denoiser generation with different $\lambda^*$.

both checkpoints share the same Stable Diffusion 1.5 as teachers. Note that the SD-turbo results are obtain from the public checkpoint [3] fine-tuned from Stable Diffusion 2.1, which is different from our teacher model.

From the comparison, we observe that our model significantly outperforms distillation-based methods including LCM and InstaFlow, and it demonstrates better diversity and quality than GAN-based SD-turbo. The visual quality is on-par with 50-step generation from the teacher model.

We show additional samples from our model on a more diverse set of prompts in Table 14 and 15.

---

[3] https://huggingface.co/stabilityai/sd-turbo

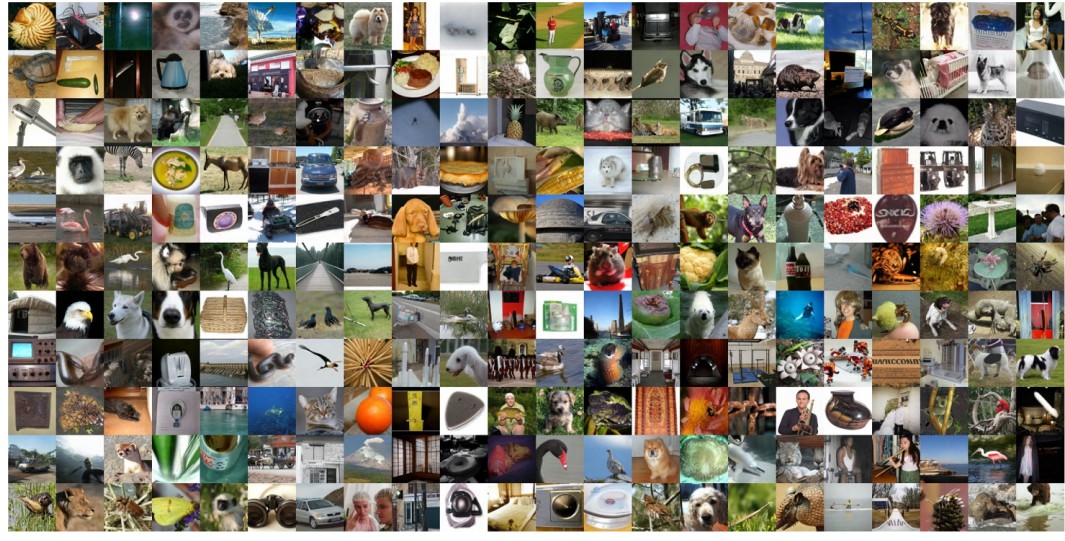

(a) ImageNet 64×64 Multi-class

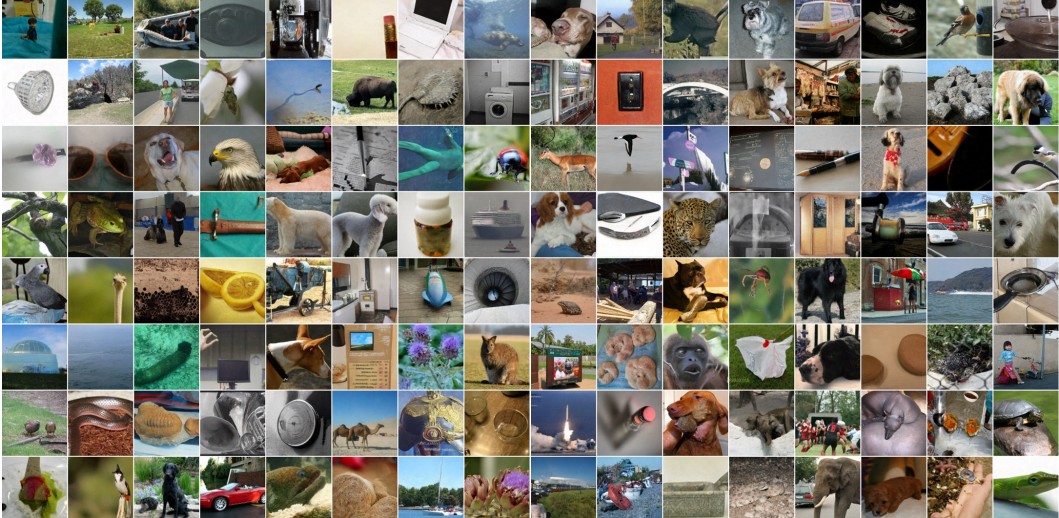

(b) ImageNet 128×128 Multi-class

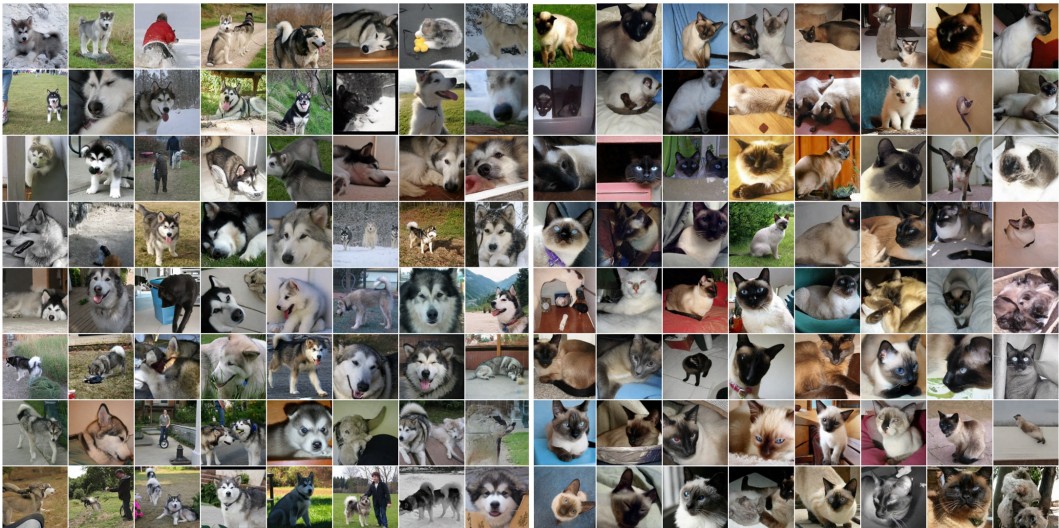

(c) ImageNet 128×128 Single-class (Left: Husky, right: Siamese)

Figure 7: Additional qualitative results for ImageNet

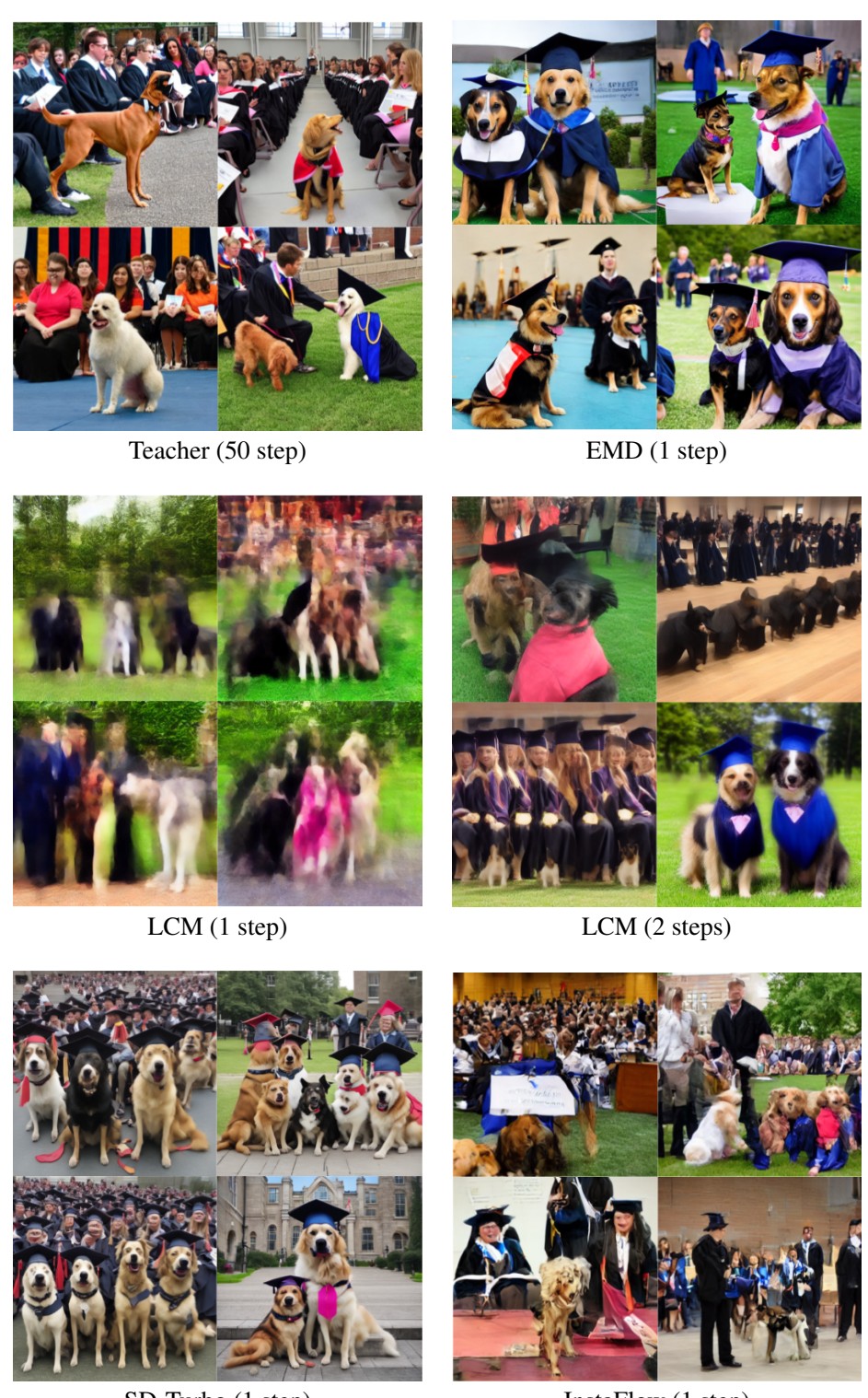

Teacher (50 step)

EMD (1 step)

LCM (1 step)

LCM (2 steps)

SD-Turbo (1 step)

InstaFlow (1 step)

Table 10: Prompt: *Dog graduation at university.*

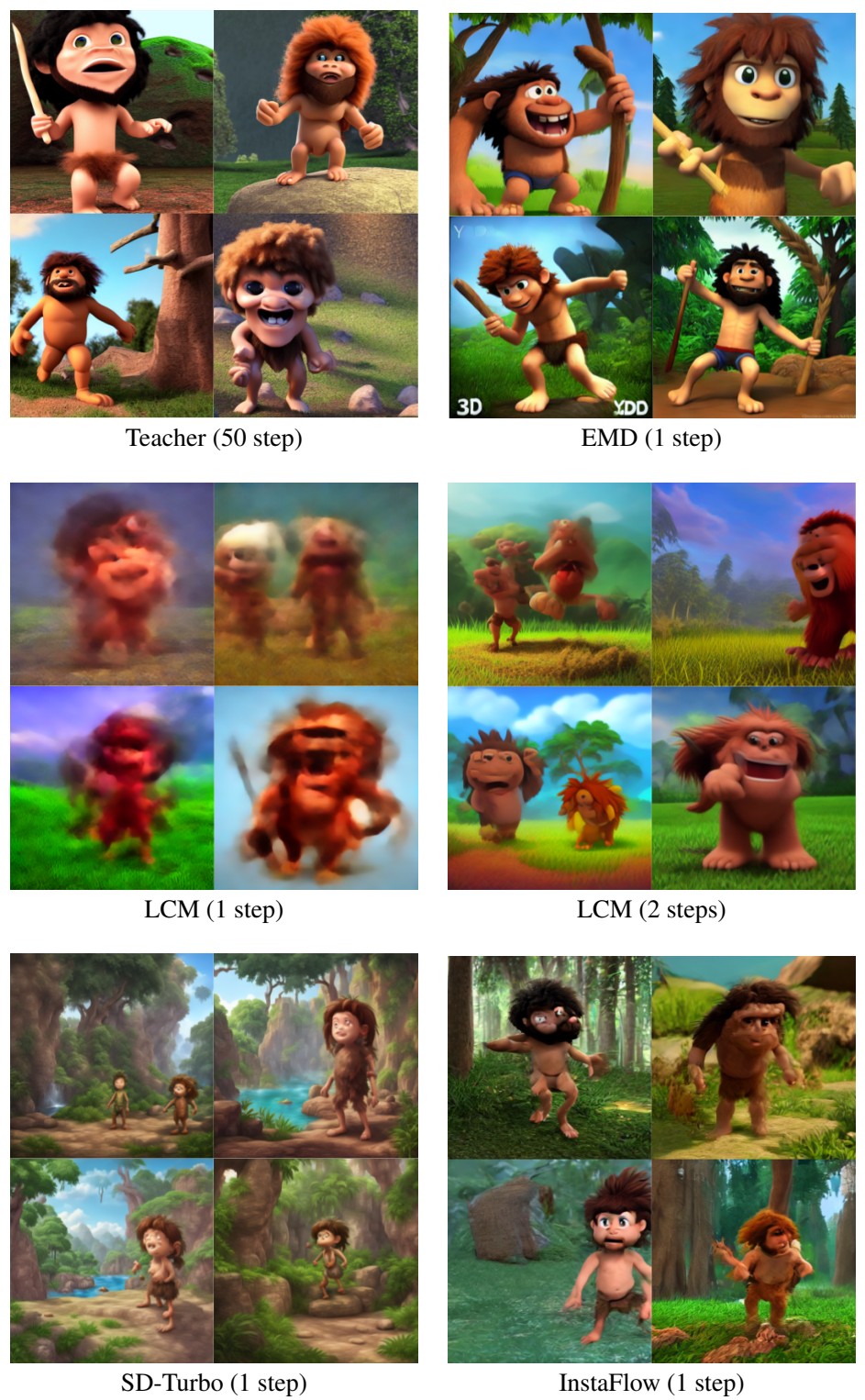

Teacher (50 step)  EMD (1 step)

LCM (1 step)  LCM (2 steps)

SD-Turbo (1 step)  InstaFlow (1 step)

Table 11: Prompt: *3D animation cinematic style young caveman kid, in its natural environment.*

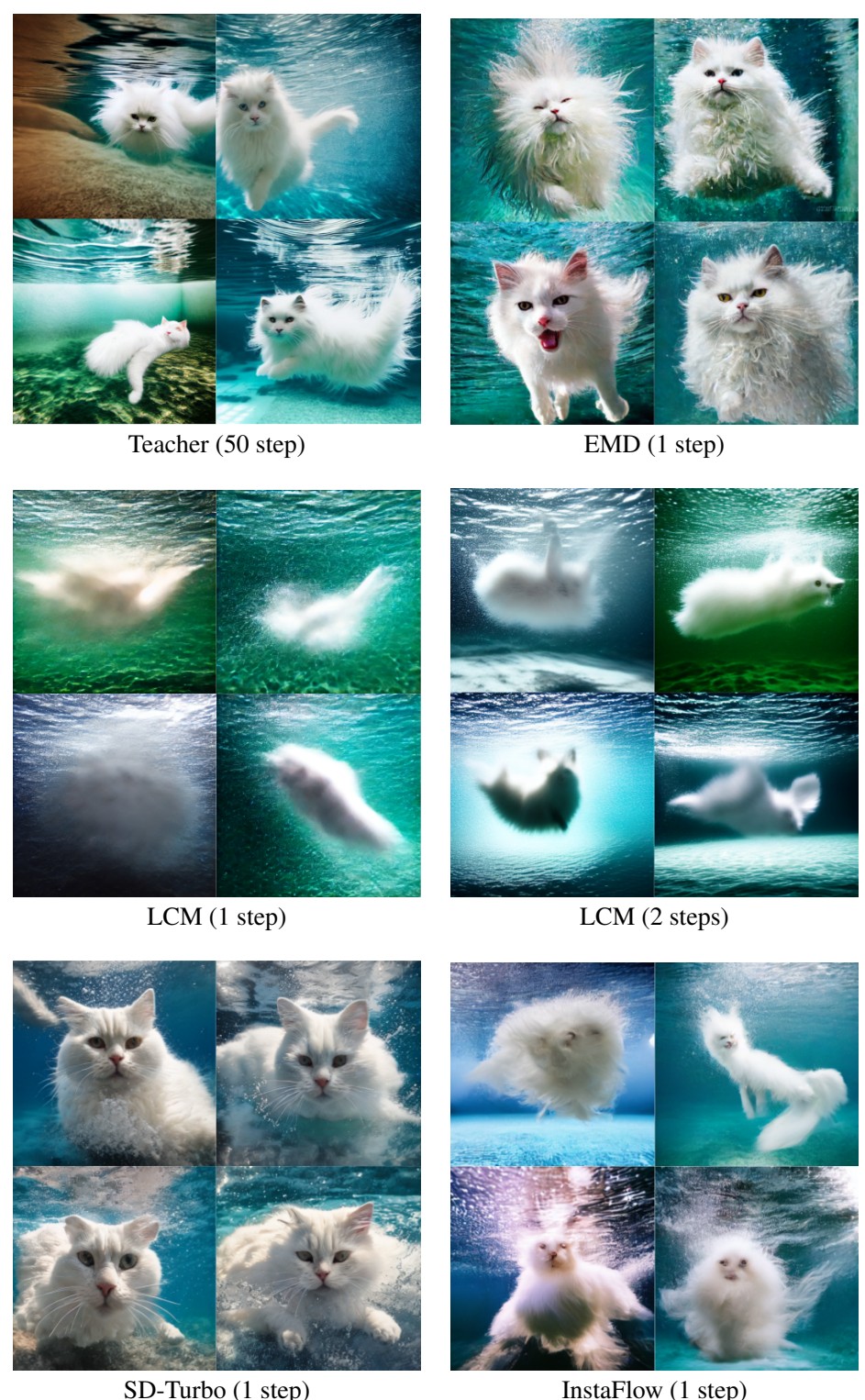

Teacher (50 step) — EMD (1 step) — LCM (1 step) — LCM (2 steps) — SD-Turbo (1 step) — InstaFlow (1 step)

Table 12: Prompt: *An underwater photo portrait of a beautiful fluffy white cat, hair floating. In a dynamic swimming pose. The sun rays filters through the water. High-angle shot. Shot on Fujifilm X.*

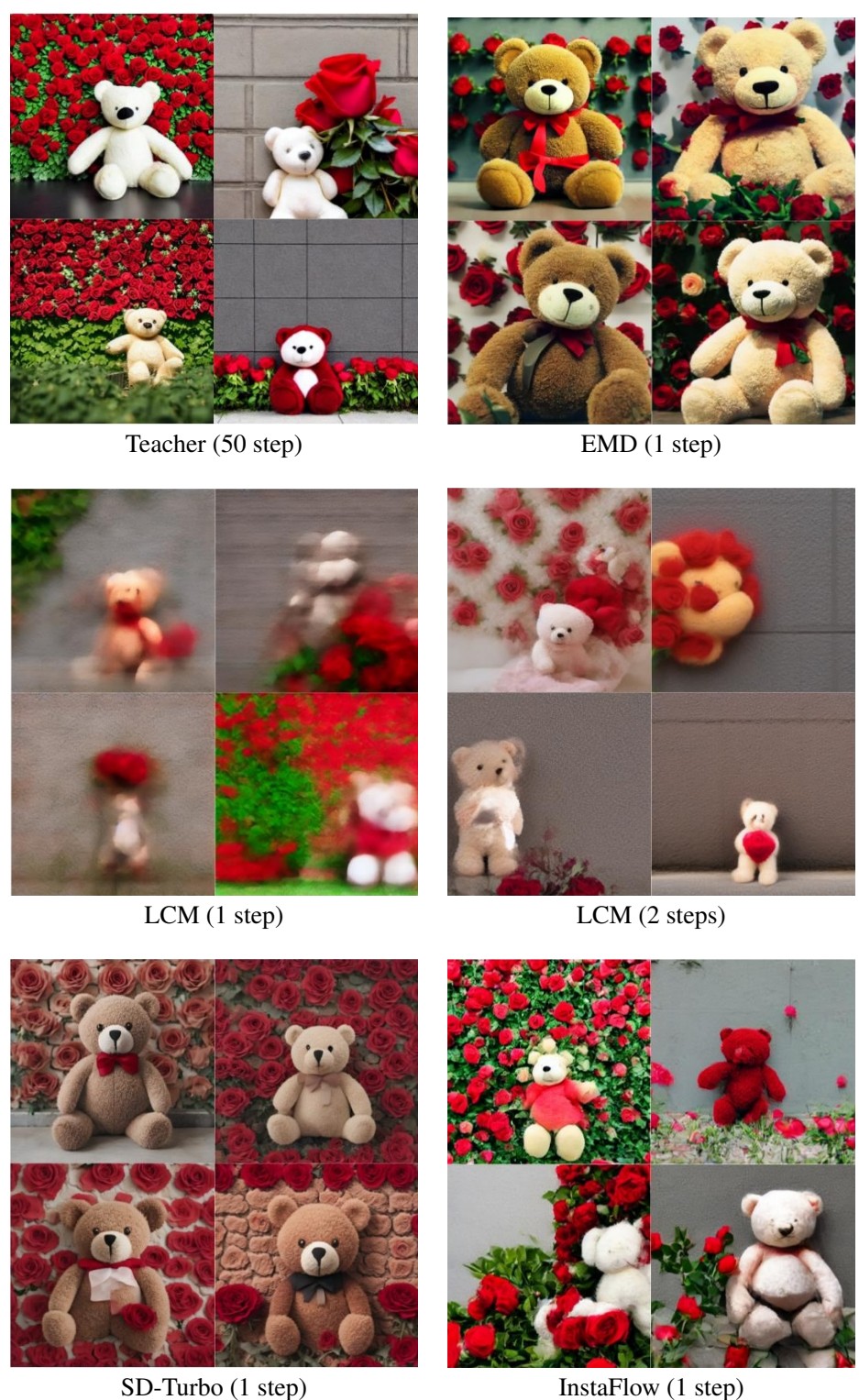

Teacher (50 step)  EMD (1 step)

LCM (1 step)  LCM (2 steps)

SD-Turbo (1 step)  InstaFlow (1 step)

Table 13: Prompt: *A minimalist Teddy bear in front of a wall of red roses.*

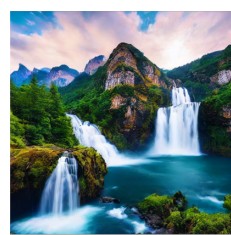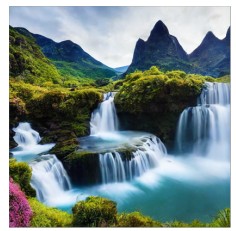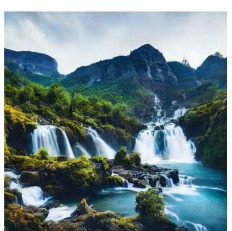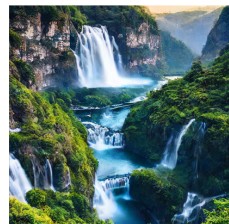

*A close-up photo of a intricate beautiful natural landscape of mountains and waterfalls.*

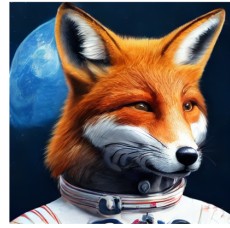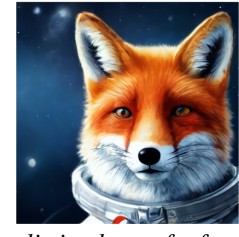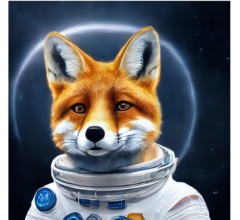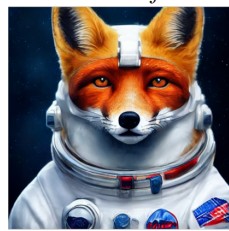

*A hyperrealistic photo of a fox astronaut; perfect face, artstation.*

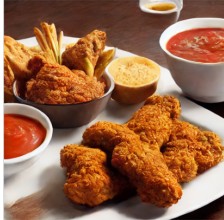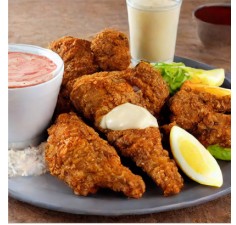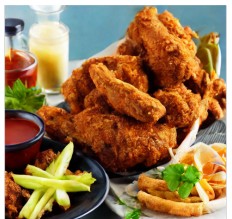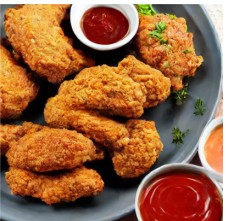

*Large plate of delicious fried chicken, with a side of dipping sauce,*
*realistic advertising photo, 4k.*

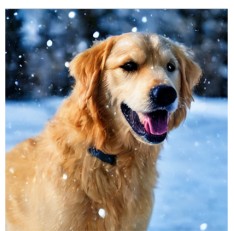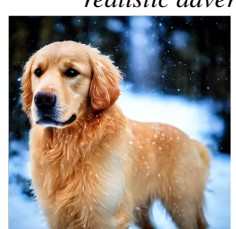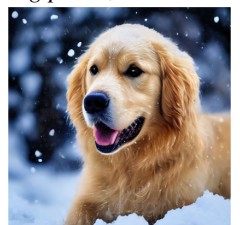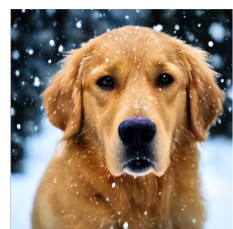

*A DSLR photo of a golden retriever in heavy snow.*

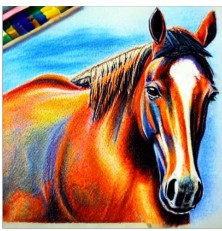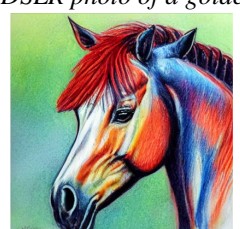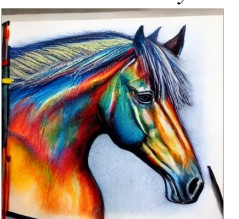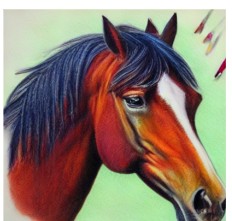

*Masterpiece color pencil drawing of a horse, bright vivid color.*

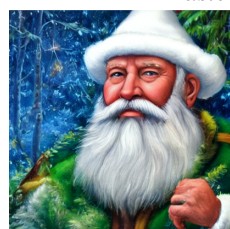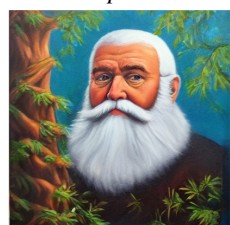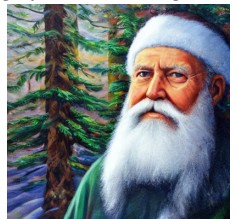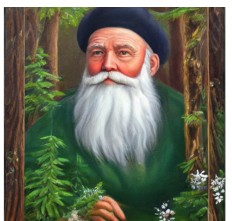

*Oil painting of a wise old man with a white beard in the enchanted and magical forest.*

Table 14: Additional qualitative results of EMD. Zoom-in for better viewing.

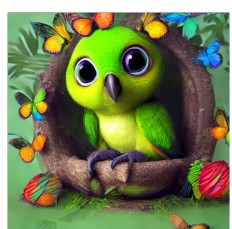 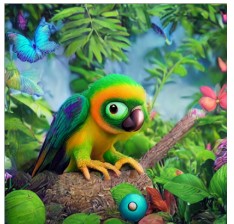 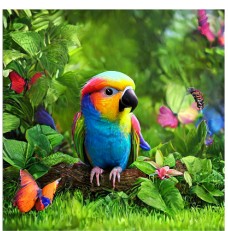 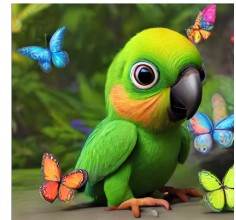

*3D render baby parrot, Chibi, adorable big eyes. In a garden with butterflies, greenery, lush whimsical and soft, magical, octane render, fairy dust.*

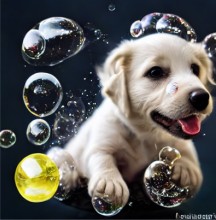 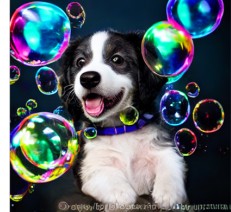 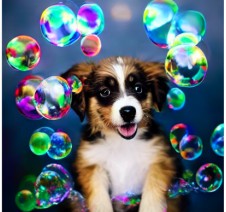 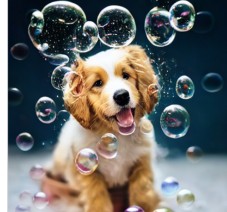

*Dreamy puppy surrounded by floating bubbles.*

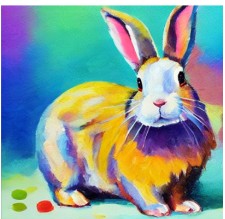 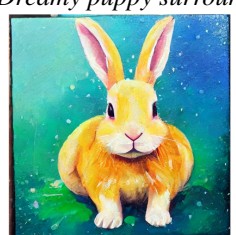 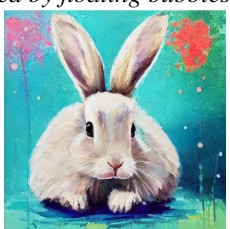 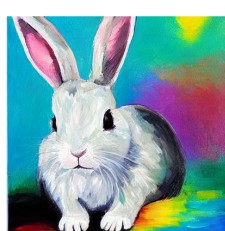

*A painting of an adorable rabbit sitting on a colorful splash.*

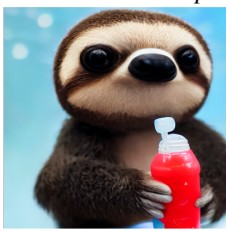 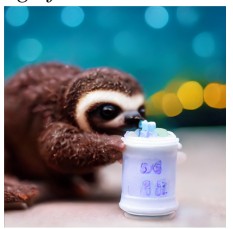 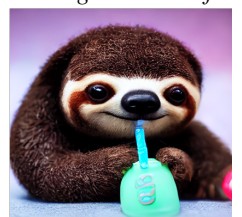 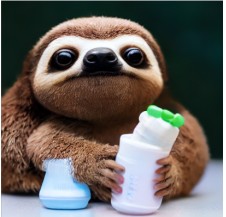

*Macro photo of a miniature toy sloth drinking a soda, shot on a light pastel cyclorama.*

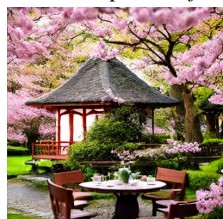 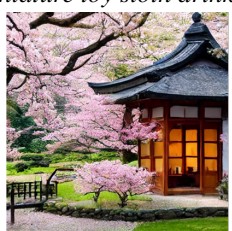 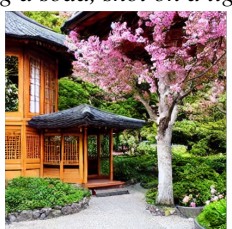 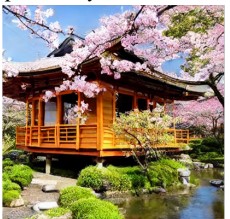

*A traditional tea house in a tranquil garden with blooming cherry blossom trees.*

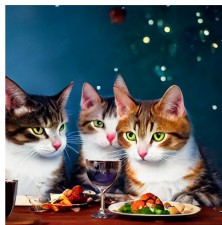 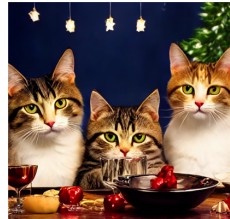 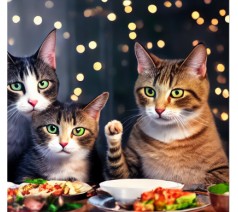 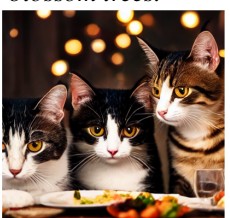

*Three cats having dinner at a table at new years eve, cinematic shot, 8k.*

Table 15: Additional qualitative results of EMD. Zoom-in for better viewing.

