# OpenReview forum: "EM Distillation for One-step Diffusion Models"
_NeurIPS.cc/2024/Conference — NeurIPS 2024 poster_

### Official Review · Reviewer_MtDz · 2024-07-13

**Soundness:** 2
**Presentation:** 2
**Contribution:** 2
**Rating:** 6
**Confidence:** 3

**Summary:**

This work proposes EM Distillation which uses the idea of expectation maximization (EM) to distill a pretrained diffusion model. Naive adaptation of EM algorithm can be computationally expensive as it requires sampling from the teacher diffusion model (which can be slow). This work proposes an alternate approach to avoid sampling from the pretrained diffusion model. The idea is to run MCMC on the joint distribution $p(x,z)$ to generate samples. To further simply Langevin dynamics, the paper uses reparametrization trick and then performs Langevin updates on Gaussian noise $\epsilon$ as well as the latents z (which come from the prior $p(z)$). As each step of Langevin dynamics adds noise, this adversely affects training due to large variance. The paper suggests getting rid of this additional noise to stabilize training (called noise cancellation in paper). The proposed method generalizes previously proposed Variational Score Distillation-based methods such as Diff-Instruct and seems to perform well on conditional image generation tasks.

**Strengths:**

1. The qualitative results for 1-step generation with EMD are impressive (Figure 6-14 in the appendix). The quantitative metrics seems comparable or better on datasets like ImageNet-64X64/128X128, and MS-COCO etc.
2. The proposed method can generate high quality images in one step. Further, the method also shows improved diversity of generated images for the same prompt. This indicates that the method indeed results in improved mode coverage.
3. I like the idea of using stochastic Langevin updates to get better mode coverage but later getting rid of the added noise to aid smoother training.

**Weaknesses:**

1. This method needs more calls to teacher model compared to the baseline methods like Diff-Instruct and DMD. The corresponding overhead associated in training, both in terms of additional training time and compute, should be discussed in the paper.
     1. The additional computational overhead needs additional clarity. For instance, from Algorithm 1 and 2, it seems that this method differentiates through the Langevin update steps. If K=16 for instance, this would mean that the computational graph  will be $16\times$ larger compared to the baseline $K=1$ like Diff-Instruct, as it would require backprop through the generator network $K$ times.
2. The paper unfortunately has some significant typos. Some equations in Section 3.2 are missing some multiplicative factors (See more in the questions below). As a result, I’m not sure if the experimental observations made from the experiments in Section 3.2 are correct. I hope the authors can clarify my questions below so that I can adjust my score accordingly.
3. This method introduces additional hyper-parameters but the corresponding ablation studies for hyper-parameter sensitivity are missing. For instance, how does the performance of EDM vary with K, the steps of MCMC, during training? Currently, the paper mostly considers two cases K=1 and K=16. It is also unclear how $t^\star$ is selected and how it is used.  It appears suddenly in the main text in Section 5.2. (See more questions below)

**Questions:**

1. The expectation in Eq 5 seems incorrect. $\epsilon$ is undefined. I think the expectation should be w.r.t $p(t), p(z), x_0 \sim g_\theta(z)$, and  $p(x_t|x_0)$ ie. $E_{p(t), p(z), x_0 \sim g_\theta(z),  x_t \sim p(x_t|x_0) }[\cdot]$.
2. There seems to be a parameter $\alpha_t$ missing in equation 7. $\nabla_z \log p_\theta(x|z) = \dfrac{\alpha_t}{\sigma_t^2}(x_t - \alpha_t g_\theta(z))^\top \nabla_\theta g_\theta(z)$. Also, why is score $\nabla_z \log p_\theta(z)$ simplified to $z$ in this equation?
3. I find line 142 confusing. $\epsilon$ is again undefined here but from Algorithm 1, it seems to be i.i.d. sample from $\mathcal{N}(0, I)$. If so, how can $\alpha g_\theta(z) + \epsilon$ be a deterministic transformation? Is $\epsilon$ always fixed for a given $z$? This seems like the reparametrization trick used in VAEs but that still doesn’t make this transformation deterministic. Also, if $p_\theta(x_t|z) = \mathcal{N}(\alpha_t g_\theta(z), \sigma_t^2I)$, then shouldn’t this transformation be $x_t = \alpha g_\theta(z) + \sigma_t \epsilon$?
4. By noise cancellation, does it mean that we collect all terms $\sqrt{2 \gamma} n^i$ from equation 16 for K steps, and then subtract it after K steps?
5. How sensititive is the training and final performance of the model to the specific choices of $K, \gamma_e$ and $\gamma_x$?
6. What is $\lambda^\star$ and how is its value determined? [Table 6, hyperparameters] How sensitive is the final performance to the choice of this hyperparameter?
7. What does $\epsilon$-correction without $z$-correction mean? Is $g(z)$ fixed for all MCMC steps and only $\epsilon$ updated? (This is used in Table 1).

**Limitations:**

The paper discusses its limitations.

---

> ### Author Rebuttal · Authors · 2024-08-07
>
> We thank the reviewer for their feedback! It is very encouraging to know the reviewer liked the idea of combining Langevin updates with noise reduction, as well as our experiment results.
>
> **[Computational overhead]**
>
> See Global Response
>
> **[Typos]**
>
> We sincerely apologize for the typos in the submitted version. Here are some clarifications:
> 1. We should have put the definition of $\epsilon\sim\mathcal{N}(0, I)$ and $x_t = \alpha_t g_\theta(z) + \sigma_t \epsilon$ earlier such that the expectation in Eq (5) is a result of reparametrization. $x = \alpha g_\theta(z) + \epsilon$ in L142 is a typo (as it misses a $\sigma$), the correct one is in L177.
> 2. In Eq (7), missing the parameter $\alpha_t$ is a typo. We have double-checked our implementation. Thanks for pointing it out.
>
> **[Noise cancellation]**
>
> Yes, you are completely right. We stated this in L521-533: “Empirically, we find book-keeping the sampled noises in the MCMC chain and canceling these noises after the loop significantly stabilizes the training of the generator network.” We will move these lines to the main text in the revised version.
>
> **[Hyperparameters of MCMC (K, stepsize)]**
>
> See Global Response for the ablation of MCMC steps.
>
> As for the step size, we found $\gamma_\epsilon\in[0.3^2, 0.4^2]$ and  $\gamma_z\in[0.003^2, 0.004^2]$ are generally good for the 3 tasks we experimented with. We reported the best configuration in the manuscript.
>
> **[t\* and  \lambda\*]**
>
> We respectfully argue that t\* does not suddenly appear in Section 5.2. In Section 3.1, L113-116, we provide an introduction of t\* under the context of the diffusion denoiser, i.e. the x-prediction function. The intuition is that by choosing the value of t\*, we choose a specific denoiser at that noise level. When parametrizing t, the log-signal-to-noise ratio $\lambda$ is more useful when designing noise schedules, a strictly monotonically decreasing function $f_\lambda$  [1]. Due to the monotonicity, $\lambda^*$ is an alternative representation for t\* that actually reflects the noise levels more directly. (During rebuttal, we find there is another typo in Table 6 and Table 7, for they reported -logSNR. We will correct them in the revision. )
>
> The pdf in the global response provides the denoiser generation at the 0th training iteration for different $\lambda^*$. When $\lambda^*=0$, the generated images are no different from Gaussian noises. When $\lambda^*=-6$, the generated images have more details than $\lambda^*=-10$. In the context of EMD, these samples help us understand the initialization of MCMC. According to our experiments, setting $\lambda^*\in[-3, -6]$ results in similar performance. For the numbers reported in the manuscript, we used the same $\lambda^*$ as the baseline Diff-Instruct on ImageNet-64 and only did a very rough grid search on ImageNet-128 and Text-to-image.
>
> [1] Kingma et al. "Variational diffusion models." NeurIPS 2021.
>
> **[What does eps-correction without z-correction mean?]**
>
> $\epsilon$-correction without $z$-correction is to fix $g(z)$ and only update $\epsilon$. It is not a theoretically rigorous algorithm. The reason we include this is that we need a baseline to show the marginal benefit of the update on $z$. The context here is that when the step sizes in $z$ and $\epsilon$ are not aligned, the performance of the joint update can be worse than the update on $\epsilon$ only. In Table 1, we showed that the reparametrized sampling eases the burden of co-adjustment of step sizes in the data and latent spaces.

---

> > ### Comment · Reviewer_MtDz · 2024-08-12
> >
> > Thanks for answering my questions. I am satisfied with the response and increasing my score.

---

> > > ### Author Response · Authors · 2024-08-12
> > >
> > > Thank you very much!

---

### Official Review · Reviewer_fdee · 2024-07-14

**Soundness:** 3
**Presentation:** 3
**Contribution:** 3
**Rating:** 6
**Confidence:** 4

**Summary:**

The paper proposes EM Distillation (EMD), a maximum likelihood-based approach that distills a diffusion model to a one-step generator model with minimal loss of perceptual quality. Notably, in EMD, the generator parameters are updated using samples from the joint distribution of the diffusion teacher prior and inferred generator latents by MCMC sampling. The method forms an extension of VSD and Diff-instruct. The empirical results are good.

**Strengths:**

- The resulting methodology becomes an extension of VSD and Diff-instruct, which is interesting and novel.
- The EMD method holds the flexibility to trade off training efficiency and final performance by adjusting the MCMC steps.
- The empirical results of the paper are strong.

**Weaknesses:**

- The noise cancellation trick is not well justified in theory, although with empirical evidence.
- It seems that there are more hyper-parameters (for the MCMC steps) to tune. I wonder about the cost of doing so. Do you have any guidance on tuning them?
- Is the trick of tuning t* used by previous works on text-to-image diffusion distillation? What is the result of doing so for prior works like DMD?

**Questions:**

See above

---

> ### Author Rebuttal · Authors · 2024-08-07
>
> Thank you for the feedback! We are very glad to see the reviewer appreciated EMD’s flexibility to trade off training efficiency and final performance by adjusting the MCMC steps. Below we respond to questions and concerns:
>
> **[Theoretical justification for noise cancellation]**
>
> We agree that there is no rigorous proof for a guaranteed variance reduction at this point. But we also want to highlight that even at worst case, noise cancellation won’t introduce biases, for canceling the noises whose mean is 0 won’t affect the mean of the gradients.
>
> **[More hyper-parameters (for the MCMC steps) to tune]**
>
> See Global Response for ablation of MCMC steps.
>
> As for the step size, we found $\gamma_\epsilon\in[0.3^2, 0.4^2]$ and  $\gamma_z\in[0.003^2, 0.004^2]$ are generally good for the 3 tasks we experimented with. We reported the best configuration in the manuscript.
>
> **[Tuning t\* for prior works on text-to-image like DMD?]**
>
> While the DMD paper didn’t report the numbers for the Diff-Instruct baseline in their setting of text-to-image, we tuned the t\* for Diff-Instruct on text-to-image and obtained results better than those reported in DMD (FID 10.96 vs 11.49). We don’t have the resources to tune DMD on text-to-image generation during the limited time of rebuttal.

---

> > ### Comment · Reviewer_fdee · 2024-08-11
> > **Thanks for the rebuttal**
> >
> > Thanks for the rebuttal. I recommend the authors clarify these things in the revision:
> >
> > 1. Regarding noise cancellation, though the mean is not affected, the variance can also significantly affect the optimization. So, more analyses are required.
> >
> > 2. What is the performance of the baselines combined with some used tricks in the paper?

---

> > > ### Author Response · Authors · 2024-08-11
> > >
> > > Thank the reviewer for this comment. We will include the following clarification in the revision:
> > >
> > > 1. While the noise cancellation technique improved performance on three distinct tasks (ImageNet-64, ImageNet-128, Text-to-image), the theoretical analysis of its variance reduction effect indeed needs more investigation.
> > > 2. As stated in L210-211 and L266-267, the EMD-1 baseline is the Diff-Instruct baseline with all the "tricks" (same $t^*$, same learning rate, etc.). We will reiterate it in the revision.

---

### Official Review · Reviewer_p5As · 2024-07-15

**Soundness:** 3
**Presentation:** 3
**Contribution:** 3
**Rating:** 6
**Confidence:** 4

**Summary:**

This paper introduces a novel distillation method for converting a diffusion process into a one-step generator. The theoretical foundation is closely tied to the Expectation Maximization (EM) algorithm. The authors aim to minimize the forward Kullback-Leibler (KL) divergence between the target generator distribution and the one-step sampler's output distribution.

This minimization is approached using an EM-like algorithm. In the expectation step, Markov Chain Monte Carlo (MCMC) sampling is used to sample from the joint distribution of noise and the generated image, followed by an MCMC correction to adjust towards the distribution of noise and the target image. The estimated gradients from this process are then used to update the generator. Simultaneously, an auxiliary diffusion model is trained to approximate the score of the output distribution.

The authors draw an interesting connection between their proposed EM distillation algorithm and previous score distillation-based methods. The final model is evaluated on both class-conditional and text-conditional image generation tasks.

**Strengths:**

S1. The theoretical framework is both novel and robust, showcasing an innovative adaptation of the EM algorithm for diffusion distillation. The paper clearly delineates the connection with previous methods based on reverse KL minimization.

S2. The writing is clear, with intuitive mathematics and excellent presentation.

S3. A wide range of design choices, such as reparameterized sampling and noise cancellation in gradient estimation, are well-founded and enhance performance.

**Weaknesses:**

W1. The performance improvement over baseline approaches (such as 1-step EMD used in VSD, DiffInstruct, DMD, Swiftbrush) is minimal, particularly for text-to-image synthesis. There appears to be a significant gap between the distilled models and the original diffusion teacher.

W2. While the forward KL divergence is theoretically mode-covering, it still has significantly worse recall compared to the teacher model or trajectory-preserving approaches like the consistency model. Could the authors comment on why this is the case?

W3. It seems that the code will not be released. While this is not a critique, providing additional details, such as pseudocode for the MCMC correction, would be helpful for reimplementation.

**Questions:**

All my concerns are detailed in the weakness section.

**Limitations:**

The authors adequately discussed the limitations.

---

> ### Author Rebuttal · Authors · 2024-08-07
>
> We thank the reviewer for their feedback. It is our honor to know the reviewer liked the framework, exposition and technical inventions in our paper.  One small correction we want to make in the summary is that in the expectation step, Markov Chain Monte Carlo (MCMC) sampling is initialized with samples from the joint distribution of noise and the generated image, as the initial sampling of z and x does not involve MCMC. We respond below to several questions and concerns:
>
> **[Performance]**
>
> We want to respectively argue that on ImageNet-64, where the metric is more reliable, EMD improves the FID significantly from 3.1 of the 1-step baseline to 2.2. For text-to-image generation, we only use the zero-shot MSCOCO FID as a proxy for the evaluation following works in the literature However, the metric might be less reliable, as the teacher model’s distribution may be different from the data from MSCOCO.
>
> Concurrent to our work, [1] studies some optimization techniques in the framework of DMD to reduce the bias in the student score model. We would like to try these techniques in future work to see if it improves the MCMC correction in EMD.
>
> [1] Yin et al. "Improved Distribution Matching Distillation for Fast Image Synthesis." arXiv 2024.
>
> **[Worse recall compared to the teacher model or trajectory-preserving approaches?]**
>
> This is a good question, which we discussed a bit in L235-236: “A larger number of Langevin steps encourages better mode coverage, likely because it approximates the mode-covering forward KL better.” The short-run MCMC sampler only produces an approximation of the posterior mean, so it is possible that there are still mode missing. EMD is better to be understood as an interpolation between mode-seeking and mode-covering KL. It achieves high perceptual quality and avoids significant mode seeking.
>
> **[Code release and pseudocode for the MCMC correction]**
>
> We plan to release code for ImageNet-64 by the camera-ready deadline. The pseudocode for MCMC correction is provided in Algorithm 2. We will be happy to polish it in the revised version.

---

> ### Comment · Reviewer_p5As · 2024-08-07
>
> I thank the authors for their response and will maintain my original accept rating. Additionally, I believe the doubling of runtime is not a significant issue, provided that it results in notable improvements in quality. Further enhancements in text-to-image quality and recall would be interesting future directions.
>
> For the pseudo code, I was referring to some pytorch-style pseudo code that can be directly transferred e.g. the one in Moco Algorithm 1 [1]. But of course, it is even better to have the full imagenet code release.
>
> [1] He, Kaiming, et al. "Momentum contrast for unsupervised visual representation learning." CVPR. 2020.

---

> > ### Author Response · Authors · 2024-08-12
> >
> > Thank you very much for your very supportive comments! We will include a pytorch-style pseudo code in the revised version.

---

### Official Review · Reviewer_3oLm · 2024-07-15

**Soundness:** 3
**Presentation:** 3
**Contribution:** 3
**Rating:** 5
**Confidence:** 4

**Summary:**

This paper introduces the EM Distillation (EMD) method, which efficiently distills diffusion models into a one-step generator model. It utilizes a maximum likelihood approach grounded in Expectation-Maximization (EM) and maintains good image generation quality. The method incorporates a reparametrized sampling scheme and a noise cancellation technique, enhancing the stability of the distillation process. EMD demonstrates good FID scores relative to existing one-step generative models on ImageNet-64 /128 and exhibits capabilities in distilling text-to-image diffusion models.

**Strengths:**

1.	The paper is well-written and easy to understand.
2.	It demonstrates the effectiveness of the proposed method on a large scale through text-to-image experiments.
3.	Although the methodology builds on existing concepts, its application of latent variable models and the EM algorithm provides a novel perspective on the distillation problem.

**Weaknesses:**

Weaknesses:

1.	The requirement for at least K steps of MCMC sampling significantly increases training costs, in contrast to other methods such as SDS, VSD, SiD and consistency distialltion (CD), which generally require only one-step sampling for distillation.
2.	While the performance with a large number of MCMC steps is robust, it still does not achieve state-of-the-art results when compared to SiD, particularly with a smaller number of steps.

Missing Comparisons:

SiD[1] derived a new distillation method originated from fisher divergence, and reach FID 1.52 on ImageNet-64x64.

[1] Zhou, Mingyuan, et al. "Score identity distillation: Exponentially fast distillation of pretrained diffusion models for one-step generation." Forty-first International Conference on Machine Learning. 2024.

**Questions:**

Questions:

1. Why modeling the joint distribution of (x, z)? Will this bring any benefits?
2. Without MCMC sampling on $z$, the loss function will be reformed to VSD loss. VSD doesn't require more than one-step MCMC. Does this mean that involving $z$ in the sampling makes the algorithm slower? Then what is the meaning of it?

**Limitations:**

The authors discussed the limitations.

---

> ### Author Rebuttal · Authors · 2024-08-07
>
> Thank you for the feedback! We feel very encouraged to receive the recognition that EMD provides a novel perspective on the distillation problem. Below we respond to your questions and stated weaknesses:
>
> **[Overhead for using MCMC in training]**
>
> See Global Response
>
> **[Comparison with SiD]**
>
> Thanks for bringing this insightful paper to our attention. We weren't aware of it at the time of submission and will definitely include this citation in the revised version. We think it is a very interesting future direction to see if the objective decomposition techniques proposed in SiD can be incorpated in EMD.
>
> **[Why modeling the joint distribution of (x, z)? Why running the “slow” MCMC sampling on (x, z)?]**
>
> When viewing the 1-step generator model as a latent-variable model, the generative modeling problem naturally becomes “learning a joint distribution of latent z and data x such that the marginal distribution of x matches”. Then it naturally leads us to the EM framework that matches the marginal distribution with mode-covering forward KL.
>
> We include the ablation of different MCMC steps K in Fig. 3(c)(d) and discuss it in L231-236. The results read that both FID and VGG-Recall show clear improvement monotonically as the number of Langevin steps increases. We also provide a possible explanation that a larger number of Langevin steps encourages better mode coverage, likely because it approximates the mode-covering forward KL better.

---

> > ### Comment · Reviewer_3oLm · 2024-08-12
> >
> > I personally don't think using MCMC with a few steps for approximating such high dimensional joint distribution is a good choice. Maybe removing the accumulated noise, which makes the denoising process more like an ODE, helps here. Comparison to consistecy trajetory model is also missing, which reaches 1.98 FID on ImageNet (vs 2.2 FID in this paper) with one-step generation. The performance wise is not stunning, and the computational cost is more than doubled. The cost issue is going to be further worse in large scale models, which will limit the scaling up of the method. Given the efforts, I increased but kept as a borderline score.

---

> > > ### Author Response · Authors · 2024-08-12
> > >
> > > Thanks for raising the score!
> > >
> > > For the MCMC, we hope the demonstration of 300 steps of update in Fig. 1 helps make it more convincing. EMD with fewer steps of updates can be viewed as approximately amortizing these long sampling chains.
> > >
> > > We will include the result of the consistency trajectory model (CTM) in the revised version. We also hope the reviewer would like to notice EMD-16's slightly better VGG-Recall (0.59 vs 0.57) albeit the FID gap.

---

### Official Review · Reviewer_eeRQ · 2024-07-15

**Soundness:** 2
**Presentation:** 2
**Contribution:** 2
**Rating:** 3
**Confidence:** 5

**Summary:**

This paper distill the diffusion model into single step generator through forward KL (mode-coverage) divergence. Apart from previous reverse KL divergence, it requires the joint samples z, x from student distribution. To do this, this paper utilize MCMC method to achieve the samples (z, x).

**Strengths:**

1. Implementing forward KL divergence is important because it can leverage the good statistical property of MLE.

2. The empirical results that compares (EMD 1 vs EMD 16) shows direct benefit of proposed method. Especially, the improvements of recall in table2 is align with the motivation of forward KL divergence.

**Weaknesses:**

1. There are two erroneous parts. First one is approximation of student score function through another neural network. Second one is discretization error of MCMC sampling. Can you justify the effects of this two errors in training both theoretical way and empirical way?


2. Training cost seems extremely expensive. This requires 1) separate approximation of student score, 2) 16 MCMC steps per iterations. Expensive costs itself is bad thing, but the paper does not analysis on the costs rigorously. You must add the portion of training costs (approximated student score training, MCMC, student loss computation, student model back-propagation) in one iteration. You must add performance per iteration (e.g. x-axis:iteration / y-axis: FID).


3. Adding all the metric of (NFE, FID, Pec, Rec, IS) for table2 and table3.


4. Forward KL divergence often fails to achieve better fidelity compared to reverse KL divergence. Did you see any similar situations? If not, you'd better discuss the reason.

**Questions:**

1. What if the MCMC steps becomes different? (e.g. 8, 4, 2)

2. Can you expand the method to f-divergence?

3. Will you release the code?

**Limitations:**

None.

---

> ### Author Rebuttal · Authors · 2024-08-07
>
> We thank the reviewer for their feedback. We appreciate the acknowledgment of our motivation in using forward KL and the corresponding validation in the ablation between EMD-16 and EMD-1. However, we would like to point out a misunderstanding in the summary “Apart from previous reverse KL divergence, it requires the joint samples z, x from student distribution”: The joint sampling is towards the target distribution of the teacher, instead of the student. Below is our point-to-point response.
>
> **[Approximation errors in student score and MCMC]**
>
> This is a valid concern and below we discuss the two errors separately.
>
> The approximation error of student score is unfortunately a universal issue in the family of distribution matching methods, including VSD, Diff-Instruct, DMD and EMD. Due to the generality of this issue, we believe it is worth dedicated research. Concurrent to our work, [1] finds that more optimization steps leads to better approximation. Another concurrent work mentioned by reviewer 3oLm, SiD, investigates a better way to decompose the distribution divergence. We are happy to mention this issue in the limitation section, provide pointers to these works in the revised version, and explore in future work if these techniques are transferable to our method.
>
> Empirically, we find that this discretization error in MCMC is not problematic, and short-run MCMC that we use in EMD can instead be motivated as interpolating between Moment-Matching Estimate and Maximum Likelihood Estimate [2]. In Fig 3cd, we empirically show that this error can be reduced by running longer MCMC chains.
>
>
> [1] Yin et al. "Improved Distribution Matching Distillation for Fast Image Synthesis." arXiv 2024.
>
> [2] Nijkamp et al. "Learning non-convergent non-persistent short-run mcmc toward energy-based model." NeurIPS 2019.
>
> **[Computation overhead]**
>
> See Global Response
>
> **[Performance per iteration (e.g. x-axis:iteration / y-axis: FID)]**
>
> We did include this curve in Fig. 3b.
>
> **[Adding all the metrics for table2 and table3]**
>
> Thanks for the suggestion. We will add the following numbers to Table 2 in the revised draft:
>
> EMD-16 Prec. 0.7559, IS 68.31, EMD-1 Prec. 0.7579, IS 62.43.
>
> For table 3, unfortunately, we don’t find baseline Prec. and Rec. to compare with.
>
> **[Forward KL divergence often fails to achieve better fidelity compared to reverse KL divergence]**
>
> This is a good question. Empirically we observe EMD still gives high per sample fidelity while alleviates the mode coverage problem of Diff-instruct that leverages reverse KL. A possible explanation is that the short-run MCMC sampler only produces an approximation of the posterior mean but is still far from fully mixing. Therefore, EMD is better to be understood as an interpolation between mode-seeking reverse and mode-covering forward KL, which maintains high perceptual quality and meanwhile avoids significant mode seeking.
>
> **[What if the MCMC steps become different? (e.g. 8, 4, 2)]**
>
> See Global Response.
>
> **[Can you expand the method to f-divergence?]**
>
> Upon the reviewer’s request, we reviewed the literature and found there is an alpha-EM framework [3] that generalizes EM to alpha-divergence, another special case of f-divergence. Happy to iterate with the reviewer on other types of f-divergence you deem to be interesting and better.
>
> [3] Matsuyama, Yasuo. "The/spl alpha/-EM algorithm: surrogate likelihood maximization using/spl alpha/-logarithmic information measures." IEEE Transactions on Information Theory 49.3 (2003)
>
> **[Will you release the code?]**
>
> We plan to release code for ImageNet-64 by the camera-ready deadline.

---

> > ### Comment · Reviewer_eeRQ · 2024-08-09
> >
> > Thank you for the response. I read all the reviews and rebuttals. In my opinion, the performance gain of EMD is not enough enduring the twice amount of computational cost. I think MCMC is not good way to implement forward KL in distillation scenario. For example, if you adopt auxiliary density ratio estimator (a.k.a. discriminator) between student and teacher, you can implement forward KL without MCMC. I want to keep my score.

---

> > > ### Author Response · Authors · 2024-08-09
> > >
> > > Thanks the reviewer for the comments. Can we ask for a clarification on how to implement forward KL with a discriminator? It would be very helpful if the reviewer would like to provide some pointers to existing works.

---

### Author Rebuttal · Authors · 2024-08-07

# Global Response #

We would like to thank all reviewers for your careful and helpful feedback! Specifically, we want to express our appreciation to reviewer eeRQ for recognizing the motivation of using forward KL, to reviewer 3oLm for identifying the novel perspective that EMD offers on the distillation problem, to reviewer p5As for the encouraging comments of our theoretical framework being novel and robust, writing being clear and intuitive, design choices being well founded and effective, to reviewer fdee for the particular interest in EMD’s flexibility to trade off training efficiency and final performance, and to reviewer MtDz for liking the core designs in our technical contribution.

We would like to clarify and address several common concerns here:

**[Ablation on MCMC steps]**

In the submitted version, we included the ablation of different MCMC steps K in Fig. 3(c)(d) and discussed it in L231-236. Both FID and VGG-Recall show clear improvement monotonically as the number of Langevin steps increases. We also provide a possible explanation for why a larger number of Langevin steps encourages better mode coverage as it approximates the mode-covering forward KL better.

**[Computation overhead]**

First, despite EMD being more expensive per training iteration compared to the baseline approach Diff-Instruct, we find the performance gain of EMD cannot be realized by simply running Diff-Instruct for the same amount of time or even longer than EMD. Second, the additional computational cost that EMD introduced is moderate even with the most expensive EMD-16 setting. See below for a detailed time analysis. Finally, for text-to-image generation, it takes EMD-8 and EMD-1 3h50min and  2h14min respectively to converge to the lowest FID.

In response to the reviewer eerQ’s, 3oLm’s and MtDz’s request, here we report some quantitative measurement of the computation overhead. Since it is challenging to time each python method’s wall-clock time in our infrastructure, we instead logged the sec/step for experiments with various algorithmic ablations on ImageNet-64.
| Algorithmic Ablation | sec/step|
|:---------------------------------|:------------------:|
| Student score matching only |  0.303 |
| Generator update for EMD-1 (joint sampling of eps and z) | 0.303 |
| Generator update for EMD-2 (joint sampling of eps and z) | 0.417 |
| Generator update for EMD-4 (joint sampling of eps and z) | 0.556 |
| Generator update for EMD-8 (joint sampling of eps and z) | 0.714 |
| Generator update for EMD-16 (joint sampling of eps and z) | 1.111 |
| EMD-16 ( student score + generator update w/ joint sampling) | 1.515 |
| Baseline Diff-Instruct (student score + generator update) | 0.703 |

So EMD-16 only doubles the wall-clock time of Diff-Instruct when taking all other overheads into account.

---

### Decision · Program_Chairs · 2024-09-25

**Decision:**

Accept (poster)

**Comment:**

This paper proposes a new approach for one step diffusion model sampling. The idea is to derive an EM style algorithm, which involves running a MCMC sampling process of the joint distribution of noise and data. This algorithm is shown to achieve competitive results in various settings. Most reviewers find the algorithm novel and interesting, and results strong, and the authors did a great job in the rebuttal period answering a lot of the questions. The remaining issues are mostly around MCMC, both its efficiency and whether it's the best way of approximating the forward KL. The AC leans towards accepting this work, primarily based on the considerations that this is a novel algorithm that works pretty well. It will be certainly nice to improve its training efficiency over other methods but that should not outweigh its contributions.